



# Drivers and impact of the seasonal variability of the organic carbon offshore transport in the Canary Upwelling System

Giulia Bonino[1,*], Elisa Lovecchio[2,*], Nicolas Gruber[3], Matthias Münnich[3], Simona Masina[1], and Doroteaciro Iovino[1]

[*]These authors contributed equally to this work.
[1]Ocean Modeling and Data Assimilation Division, Centro Euro-Mediterraneo sui Cambiamenti Climatici, Bologna, Italy.
[2]Ocean BioGeosciences, National Oceanography Centre (NOC), Southampton, U.K.
[3]Environmental Physics, Institute of Biogeochemistry and Pollutant Dynamics, ETH Zurich, Zürich, Switzerland

**Correspondence:** Giulia Bonino (giulia.bonino@cmcc.it), Elisa Lovecchio (elisa.lovecchio@noc.ac.uk)

**Abstract.** The Canary Upwelling System (CanUS) is a productive coastal region characterized by strong seasonality and an intense offshore transport of organic carbon ($C_{org}$) to the adjacent oligotrophic offshore waters. There, the respiration of this $C_{org}$ substantially modifies net community production (NCP). While this transport and the resulting coupling of the biogeochemistry between the coastal and open ocean has been well studied in the annual mean, the temporal variability, and especially its

seasonality has not yet been investigated. Here, we fill this gap, and determine the seasonal variability of the offshore transport of $C_{org}$, its mesoscale component, latitudinal differences, and the underlying physical and biological drivers. To this end, we employ the Regional Ocean Modeling System (ROMS) coupled to a nutrient, phytoplankton, zooplankton, and detritus (NPZD) ecosystem model. Our results reveal the importance of the mesoscale fluxes and of the upwelling processes (coastal upwelling and Ekman pumping) in modulating the seasonal variation of the offshore $C_{org}$ transport. We find that the region surrounding

Cape Blanc (21 °N) hosts the most intense $C_{org}$ offshore flux in every season, linked to the persistent, and far reaching Cape Blanc filament. Coastal upwelling filaments dominate the seasonality of the total offshore flux up to 100 km from the coast, contributing in every season season at least 80 % to the total flux. The seasonality of the upwelling modulates the offshore $C_{org}$ seasonality hundreds of km from the CanUS coast via lateral redistribution of nearshore production. North of 24.5 °N, the sharp summer-fall peak of coastal upwelling results in an export of more than 30 % of the coastal $C_{org}$ at the 100 km offshore

due to a combination of intensified nearshore production and offshore fluxes. To the south, the less pronounced upwelling seasonality regulates an overall larger, but farther-reaching and less seasonally varying lateral flux, which exports between 60 and 90 % of the coastal production more than 100 km offshore. Overall, we show that the temporal variability of nearshore processes impacts the variability of $C_{org}$ and NCP hundreds of km offshore from the CanUS coast via the offshore transport of the nearshore production.

# 1   Introduction

Eastern Boundary Upwelling Systems (EBUS) are coastal regions of immense socio-economic value for human communities, as they constitute some of the most important fisheries yields worldwide, fueled by the high productivity that characterizes





these systems (Pauly and Christensen, 1995; Chavez and Toggweiler, 1995). The high nearshore productivity of these regions
is sustained by the upwelling of nutrient-rich waters, primarily impacting the first 100 km from the coast (Carr, 2002; Mackas
et al., 2006). The main driver of this upwelling are trade winds blowing towards the equator which, combined with the Coriolis
effect, induce an offshore Ekman transport of the near surface waters. This forces a surfacing of thermocline waters along the
coast and a net upward transport, i.e., upwelling (Carr and Kearns, 2003). Additional upwelling is induced by curl of wind
stress that occurs in many offshore regions of the EBUS (Messié et al., 2009).

Both the large-scale circulation and the coastal upwelling in EBUS are characterized by substantial temporal variability. The
temporal scales vary from the decadal and interannual time scales associated with phenomena such as the Altantic Multidecadal
Oscillation (AMO) and the El Niño Southern Oscillation (ENSO) to weekly or daily variations associated with short-lived
wind anomalies (Bonino et al., 2019a; Jacox et al., 2015; Desbiolles et al., 2014). These processes induce strong modulations
of biological production, biogeochemical activity and ecosystem structure and composition (Pradhan et al., 2006; Rykaczewski
and Checkley, 2008; Frischknecht et al.). Among the four major EBUS, the Canary Upwelling System (CanUS) is characterized
by the most intense seasonal variations, which drive the majority of the observed physical and biogeochemical fluctuations in
the region (Chavez and Messié, 2009).

The CanUS is located along the northwestern African coast and constitutes the eastern boundary of the North Atlantic
subtropical and tropical circulation (Pelegrí and Peña-Izquierdo, 2015; Mittelstaedt, 1991). A combination of elevated light
and nutrient availability, high iron supply, and long residence time of the upwelled waters in the nearshore areas makes the
CanUS the second most productive EBUS (Lachkar and Gruber, 2011; Carr, 2001). Together with a complex coastal pattern of
capes and embayments, the CanUS is characterized by substantial latitudinal variations in coastal inclination and topography,
which allow us to define zonal subregions that are distinct in terms of seasonality of upwelling, biological production, and
circulation (Pelegrí and Peña-Izquierdo, 2015; Pelegrí and Benazzouz, 2015).

The northern portion of the CanUS, located roughly between 25°N and 32 °N, is dominated by the alongshore southward
flow of the Canary Current (CC) and the Canary Upwelling Current (CUC) (Figure 1), which constitute the easternmost
component of the North Atlantic Subtropical Gyre (NASG) (Pelegrí and Peña-Izquierdo, 2015; Arístegui et al., 2009). The CC
is characterized by pronounced seasonal variability, including a summer intensification and a temporary reversal in fall (Mason
et al., 2011; Hernández-Guerra et al., 2002). The northern CanUS is home to multiple semi-recurrent upwelling filaments
associated with the many capes. This region is also characterized by intense mesoscale activity in the form of coastal and
island generated eddies (Sangrà et al., 2009; Garcìa-Muñoz et al., 2004). At these latitudes, the coastally confined upwelling
signature in temperature and surface chlorophyll is characterized by a sharp peak during late summer to early fall. Offshore, the
negative wind stress curl deepens the nutricline in the open waters (Lathuilière et al., 2008; Pelegrí and Benazzouz, 2015). The
combination of these two processes gives rise to very strong offshore gradients in tracer concentrations of carbon and nutrients
(Arístegui et al., 2009).

South of 25 °N), we find the convergence zone of the southward flowing CC and of the northward flowing Mauritanian
Current (MC), the latter being characterized by a summer-fall intensification (Lázaro et al., 2005). The two currents merge and
detach from the coastline at Cape Blanc (21 °N), continuing offshore along the Cape Verde frontal zone (Pelegrí and Peña-



Izquierdo, 2015; Arístegui et al., 2009). Associated with this offshore flow is the near-permanent Cape Blanc filament, which stretches up to 700 km into the open waters (Gabric et al., 1993; Ohde et al., 2015). The size of this Cape Blanc filament is

enormous, making it one of the biggest filaments among all EBUS. Cape Blanc also defines the boundary between the northern portion of the coastline (> 21 °N) characterized by a slight east-west tilt and nearly continuous upwelling, and the southern (< 21 °N) north-south oriented coastline characterized by spring-intensified upwelling (Benazzouz et al., 2014). While the tropical circulation found south of Cape Blanc is relatively weak with small and highly seasonal coastal filaments (Peña-Izquierdo et al., 2012; Menna et al., 2016), high upwelled nutrient concentrations and a widespread positive windstress curl signature fuel high

productivity both at the coast and offshore (Arístegui et al., 2006).

  Owing to the simultaneous presence of cross-shore mass fluxes and of tracer gradients, EBUS laterally export large amounts of organic and inorganic material properties to the adjacent oligotrophic waters (Nagai et al., 2015; Lachkar and Gruber, 2011; Amos et al., 2019). Among these fluxes, the offshore transport of coastally-produced organic carbon ($C_{org}$) is especially important, as this might help to explain the purported net heterotrophy of some oligotrophic open waters that are located

adjacent to some of these EBUS (Burd et al., 2010; Arístegui et al., 2003; Pelegrí et al., 2005). For example, the CanUS was shown to transport more than one third of its coastal net community production (NCP) toward the offshore, reaching as far as 2000 km from the coast (Lovecchio et al., 2017). This enhanced supply of $C_{org}$ to the North Atlantic Subtropical Gyre increases respiration there, pushing vertically integrated net community production to become negative. Over a year, narrow coastal filaments are, on average, responsible for about 80 % of this offshore flux at 100 km from the coast, while coastally

generated mesoscale eddies dominate this transport further offshore (Lovecchio et al., 2018).

  In contrast to the relatively well studied annual mean offshore transport, substantially less is known about its seasonal nature. Given the intensity of this transport, it is very likely that the strong seasonal variability of coastal upwelling and production is readily transferred towards the open ocean, albeit with some temporal delay. There, and especially in the nutrient deprived subtropical gyres, the seasonality of the lateral supply may be rather critical, especially when this lateral supply occurs during

the summer, when the nutrient deficiency is most acute. In this paper, we quantify the seasonal variability of the offshore flux of $C_{org}$ in the CanUS. In particular, we analyze the relative role of the mean and mesoscale circulation in each season, we study the role of upwelling seasonality as physical drivers of the offshore fluxes and we discuss the impact of these offshore flux changes on the ecosystem of the open waters.

## 2 Methods

### 2.1 Model setup and output

Our analysis of the seasonality of the offshore transport in the CanUS is based on the results from a CanUS model simulation undertaken and described by Lovecchio et al. (2018). This simulation used the UCLA-ETH version of the Regional Ocean Modeling System (ROMS) (Shchepetkin and McWilliams, 2005) coupled with a Nutrient Phytoplankton Zooplankton Detritus (NPZD) ecosystem model (Gruber et al., 2006). This coupled model was run with monthly climatological forcing derived

from ERA-Interim (Dee et al., 2011) on an Atlantic telescopic grid. This grid combines a full Atlantic basin perspective with





a mesoscale-resolving resolution in the region of study, achieved through a strong grid refinement towards the north-western African coast. We refer to Lovecchio et al. (2017) and Lovecchio et al. (2018) for further details on the model, on its forcing, and for providing a careful evaluation against a large range of observational constraints.

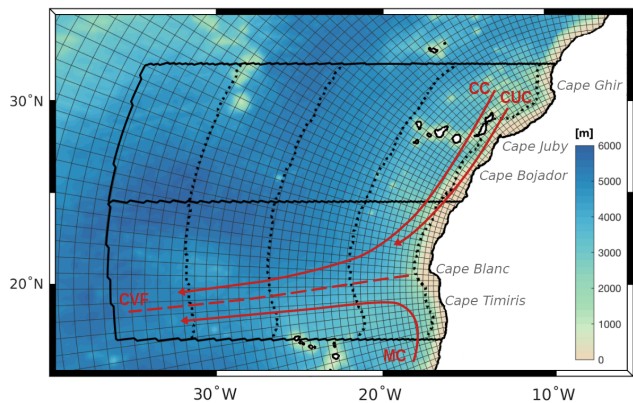

**Figure 1.** Atlantic telescopic grid zoomed around the region of study with every 5th grid line plotted. Black solid contours highlight the northern and central subregions of study. Black dotted lines indicate the offshore boundaries at 100 km, 500 km, 1000 km and 1500 km from the coast. Red lines provide a simplified scheme of the mean regional pattern of currents: Canary Current (CC), Canary Upwelling Current (CUC), Mauritanitan Current (MC) and Cape Verde Front (CVF). Shading: depth of topography [meters].

The simulation we are analyzing was run for 53 years, of which we use the last 24 years for our analyses. We use data in the form of 2-day means. On top of the necessary state variables, i.e., temperature, sea surface height (SSH), velocity fields and $C_{org}$ concentration, the model was setup to output also net community production (NCP) calculated at run time for each grid cell from total biological production minus all respiration fluxes. The 2D field of the lateral fluxes of $C_{org}$ was calculated from the model output as the product of the horizontal velocities and the tracer concentration.

In our employed NPZD model, $C_{org}$ corresponds to the sum of non-sinking zooplankton, sinking phytoplankton, and a small detritus pool that sinks slowly and a large detritus pool that sinks fast. No dissolved organic pool (DOC) is included in the model. For the purpose of our model evaluation and analysis, we define the four seasons as follows: spring is the mean of March, April, and May (MAM), summer is the mean of June, July, and August (JJA), fall is the mean of September, October, and November (SON), and winter is the mean of December, January, and February (DJF). Since one year of model output consists in 360 days, we define the unit "season" (seas) as 90 days of simulation.

## 2.2 Region of study

While we analyze the entire CanUS system from 17°N and 32°N, we focus our study on the central and northern subregions (CSR and NSR, respectively), with the border located at 24.5°N. The central subregion thus extends from 17°N to 24.5°N] and the northern subregion from 24.5°N to 32°N] (Figure 1). This correspond to the two homonym subregions studied in Lovecchio et al. (2018). Our main motivation for restricting our analyses on the CSR and NSR is that the model performs well





in these two regions, especially compared to the southern CanUS subregion (<17°N) where the model biases tend to be much
larger (Lovecchio et al., 2017, 2018).

We study tracer concentrations and fluxes from the Northwest African coast out to an offshore distance of 2000 km, corre-
sponding to the middle of the North Atlantic Gyre. We divide the offshore region of the upwelling system in five bands having
the following boundaries defined as isolines of offshore distance from the CanUS coast: 1. [0, 100 km] (offshore band directly

impacted by the coastal upwelling); 2. [100 km, 500 km]; 3. [500 km, 1000 km]; 4. [1000 km, 1500 km]; 5. [1500 km, 2000
km]. In the region of study, up to 2000 km offshore, the model grid has a resolution of between 4.7 km and 19 km, with the
highest resolution found at the upwelling coast.

### 2.3 Upwelling Estimation

To diagnose the potential drivers of the flux variability, we estimate the magnitudes of the offshore coastal transport ($U$) and of

Ekman pumping ($w_E$) from the forcing. The offshore coastal transport is obtained from the "classical" Ekman transport:

$$U = \frac{\tau_{as}}{f\rho} \tag{1}$$

where $\tau_{as}$ is the alongshore wind stress, $\rho$ is the density of sea water, and $f$ is the Coriolis parameter. $U$ ($\mathrm{m^2/s}$ or $\mathrm{m^3/s*m}$
of coast) is calculated as result of the mean $\tau_{as}$ in the first 100km from the coast in order to be consistent with the calculated
offshore fluxes. Then, it is integrated along the coast in order to obtain the vertical transport. $U$ units are $\mathrm{m^3/s}$ or Sverdrup

(Sv) (1 Sv = $10^6 \mathrm{m^3/s}$). Ekman pumping is estimated from:

$$w_E = \frac{\mathrm{curl}(\tau)}{f\rho} \tag{2}$$

where $\mathrm{curl}(\tau)$ is the wind stress curl, $\rho$ is the density of sea water, and $f$ is the Coriolis parameter. $w_E$ (m/s) is integrated
offshore in the first 100km (to obtain $\mathrm{m^2/s}$ or $\mathrm{m^3/s*m}$ of coast) and along the coast in order to obtain the vertical transport
($w_E$ in $\mathrm{m^3/s}$ or Sv). Following Messié et al. (2009), all negative values are set to 0 before integrating $w_E$ (m/s). This is done

to reflect an asymmetry in the transport of nutrients. When the Ekman pumping is positive (upward), nutrients are brought
from the thermocline to the surface and stimulate biological production. In contrast, when the Ekman pumping is negative
(downward), essentially no nutrients are transported downwards, since most of them have already been consumed.

### 2.4 Eddy and filament budgets and fluxes

The contribution of mesoscale eddies and upwelling filaments to the $C_{org}$ budget and fluxes is calculated using the same

structure identification algorithms as those employed in Lovecchio et al. (2018). Cyclonic eddies (CE) and anticyclonic eddies
(AE) are separately found using the SSH-based identification algorithm developed by Faghmous et al. (2015). This algorithm
retrieves the exact contour of each eddy at each time step according to the shape of SSH. At each time step, the CE and
AE masks are defined as 2D fields of zeros and ones, with ones corresponding to the surfaces of the retrieved eddies on the
grid. Upwelling filaments are found through the use of the sea surface temperature (SST) based identification algorithm fully

described and evaluated in the supplement to Lovecchio et al. (2018). In analogy with the eddy identification routine, this





algorithm builds a zero-one filament mask for each time step, with the ones identifying the filaments. The upwelling filament identification algorithm reads at each time step the locations of the previously identified eddies (both CE and AE) and excludes their surfaces from the filament mask. This makes sure that the filament and eddy masks never overlap. At each time step, the area that is not covered by either the eddy field or the filament field is defined as non-filament-non-eddy (NF-NE) field.

Although the algorithms identify the structures at the surface only, we assume that the eddies and filaments have a vertical prismatic structure, i.e., they occupy the same $i,j$ grid points at each depth. We consider this a good approximation as we only focus on fluxes and budgets in the euphotic layer (defined as the first 100 m depth). Fluxes and budgets for each type of structure are calculated by multiplying at each time step the eddy and filament masks by the 2D fields of concentrations or fluxes integrated vertically over the euphotic layer depth.

## 3   Evaluation

We summarize here the main relevant findings of the model evaluation for the CanUS in the region [17°N, 32°N]. We refer the reader to Lovecchio et al. (2017) and Lovecchio et al. (2018) for additional details.

In the annual mean, the model-observation misfits (biases) of SST and sea surface salinity (SSS) are smaller than 0.75 °C and 0.2, respectively. The regional pattern of currents including the alongshore flowing Canary Current and Mauritanitan

Current and the Cape Verde front are well reproduced by the model. The regional pattern and offshore gradient of net primary production (NPP) as well as surface POC (S-POC) correlate well with satellite derived estimates (correlation > 0.7) even though the model is biased slightly low at latitudes < 25 °N.

The model represents the observed seasonal variability of currents and biological activity in the region of study reasonably well (Lovecchio et al., 2017). Modeled horizontal velocities in each season reproduce in both magnitude and pattern those

obtained from drifter data (Lumpkin and Johnson, 2013), with an intensification of the Canary Current in summer and a more intense Mauritanitan Current circulation in fall and winter. The Ekman offshore transport is maximum in summer north of Cape Blanc (21 °N) and in winter-spring south of it. Biological activity also follows such a seasonal pattern, with coastal NPP showing a peak in summer and spring north and south of 21 °N, respectively. As expected, between 21 °N and 25 °N, in the proximity of Cape Blanc, coastal NPP remains quite high in all seasons compared to the surrounding latitudes. The absolute

value of NPP is biased low when compared to satellite derived products such as SeaWiFS VGPM and CbPM (Behrenfeld and Falkowski, 1997; Westberry et al., 2008), reaching respectively 1/3 and 2/3 of their magnitude. However, its pattern correlates well with SeaWiFS VGPM (0.8) and CbPM (0.7) in all seasons, giving us confidence that the relative impact of the $C_{org}$ fluxes on the offshore biological activity is well represented in the model. Absolute $C_{org}$ fluxes, however, may be biased low. Further discussion of these strengths and limitations are provided in the model evaluation and discussion of Lovecchio et al. (2017).

Both the modeled turbulent kinetic energy and the standard deviation of SSH are especially close in both pattern and magnitude to the ones derived from the AVISO satellite products (Maheu et al., 2014), confirming that the model provides a good representation of the highly variable and small scale flow in the region (Lovecchio et al., 2018). In the region of study, differences between the modeled and the satellite-derived turbulent kinetic energy consist mostly in higher modeled values in the





nearshore, likely due to small scale coastal filaments that are resolved by the model but are not captured by the lower resolution

satellite data. The regional pattern of large eddy (> 50 km diameter) density retrieved by the identification algorithm (Faghmous et al., 2015) in model and satellite data are also especially similar in both offshore gradient and absolute value, with peaks in the eddy density in the proximity of the coast and of the Canary archipelago. The portion of time occupied by filaments for each grid point according to our identification algorithm is highest in correspondence of capes that are known to be associated to recurrent upwelling filaments. We therefore expect our filament field to reproduce quite well the known regional pattern of

these structures.

## 4 Results

### 4.1 Seasonality of the total offshore transport of $C_{org}$

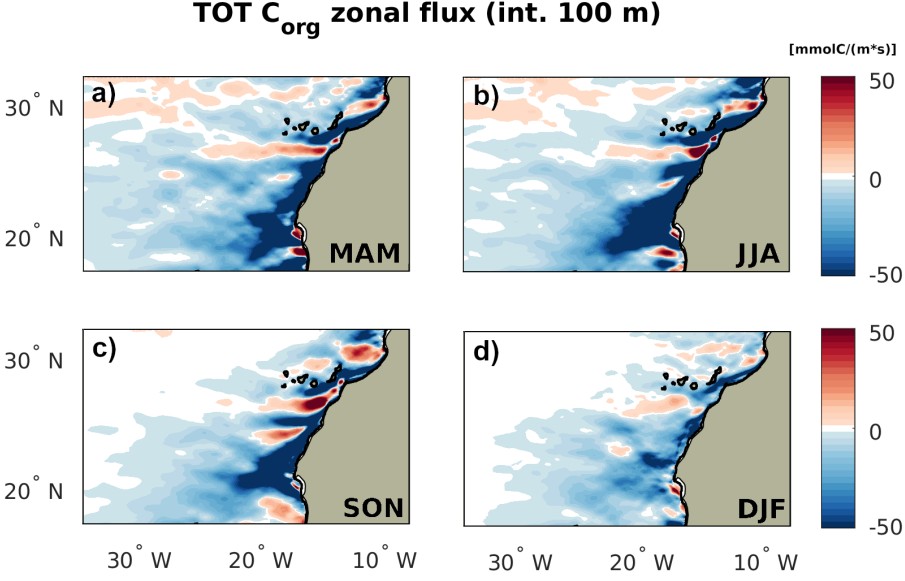

**Figure 2.** Seasonality of the total zonal transport of $C_{org}$ by season, positive indicating eastward. The flux is integrated over the top 100 m depth. Subpanels: a) spring (MAM), b) summer (JJA), c) fall (SON), d) winter (DJF).

The model simulated $C_{org}$ offshore flux, marked by a negative (westward) signature of the 100 m integrated zonal flux (Figure 2), varies seasonally in offshore extension and intensity. Summer is characterized by the most intense offshore flux in

the nearshore at all latitudes. The flux reaches farther into the open NASG in spring and summer, when it is clearly visible up to the offshore boundary of our analysis domain, at about 2000 km from the Northwest African coast.

In fall and winter, the $C_{org}$ offshore flux has a more limited extent at all latitudes, with the CSR maintaining the farthest reaching offshore flux. The intensity of the offshore transport close to the coast remains high in fall. Winter shows the weakest lateral $C_{org}$ redistribution also close to the coast at all latitudes. This reduced signal is a result of a combination of weaker





currents and lower near-surface $C_{org}$ concentrations, the latter being especially small offshore in the NSR (see Appendix: Figure B1).

The most intense offshore signature in the flux in all seasons is found in the nearshore CSR, associated with the Cape Blanc filament. The contribution of this hotspot of offshore $C_{org}$ flux is highest in summer and significantly reduced (even though still dominant) in winter. Positive and negative zonal flux striations emerge in the NSR in every season. These may be linked to the

presence of recurrent mesoscale eddies forming in the proximity of the Canary Islands (Sangrà et al., 2009).

### 4.2 Seasonality of the mesoscale and filament offshore transport

The mesoscale eddy and filament components of the $C_{org}$ zonal transport not only have different offshore extensions and signatures but also a different seasonality (Figure 3). The intense filament transport only spans about half of our analysis domain (roughly reaching 1000 km offshore),while the edddy transport is weaker, but more far-reaching. In every season the

filament flux is dominated by a negative (and therefore offshore-directed) $C_{org}$ flux, with only a minor contribution to the onshore $C_{org}$ recirculation. On the contrary, in every season the mesoscale eddy flux is characterized by striations, in the form of alternate positive and negative bands of zonal $C_{org}$ transport.

The filament transport has a strong seasonal character. In the nearshore, the filament transport is maximum in summer and very intense in fall. In these two seasons, this flux component also shows its maximum offshore extension at every latitude

and especially in the CSR around Cape Blanc. Spring sees a smoothing of the zonal gradients in the filament flux, with a more homogeneous intensity at all latitudes, especially in the nearshore. In winter, the filament transport reaches its minimum intensity, barely exceeding that of the eddy flux.

The total (cyclonic + anticyclonic) eddy contribution to the $C_{org}$ offshore flux has a very similar intensity in all seasons, with only a moderate intensification in the nearshore in summer and fall in the NSR and in spring and summer in the CSR. In terms

of the offshore extension, the eddy flux reaches its maximum in spring, when also the $C_{org}$ distribution reaches its maximum value in the offshore waters.

Across the CanUS as a whole, the relative contributions of eddies and filaments remain similar in all seasons (Figure 4). The filament flux is always the dominant flux in the nearshore, often exceeding the total offshore transport between 100 km and 200 km offshore. This is possible when the non-filament-non-eddy flux is directed onshore and therefore has a negative contribution

to the total offshore flux. The maximum relative contribution to the total flux by filaments both in terms of maximum flux share and in terms of offshore extension is found in fall, when both eddy and NF-NE fluxes are weaker. In fact, even though filaments are intense in summer, this season is also characterized by a widespread peak of the upwelling, triggered by the mean (largely NE-NF) offshore Ekman transport.

In general, the CE flux exceeds the AE flux. The CE flux contribution constitutes the largest mesoscale eddy contribution to

the total flux in all seasons and ranges mostly between 20 % and 40 % of the total, with a maximum in spring. The weaker AE flux contribution ranges between 0 and 20 % of the total offshore transport, with a small (< 5 %) relative contribution in winter and a negligible (zero) contribution in fall.



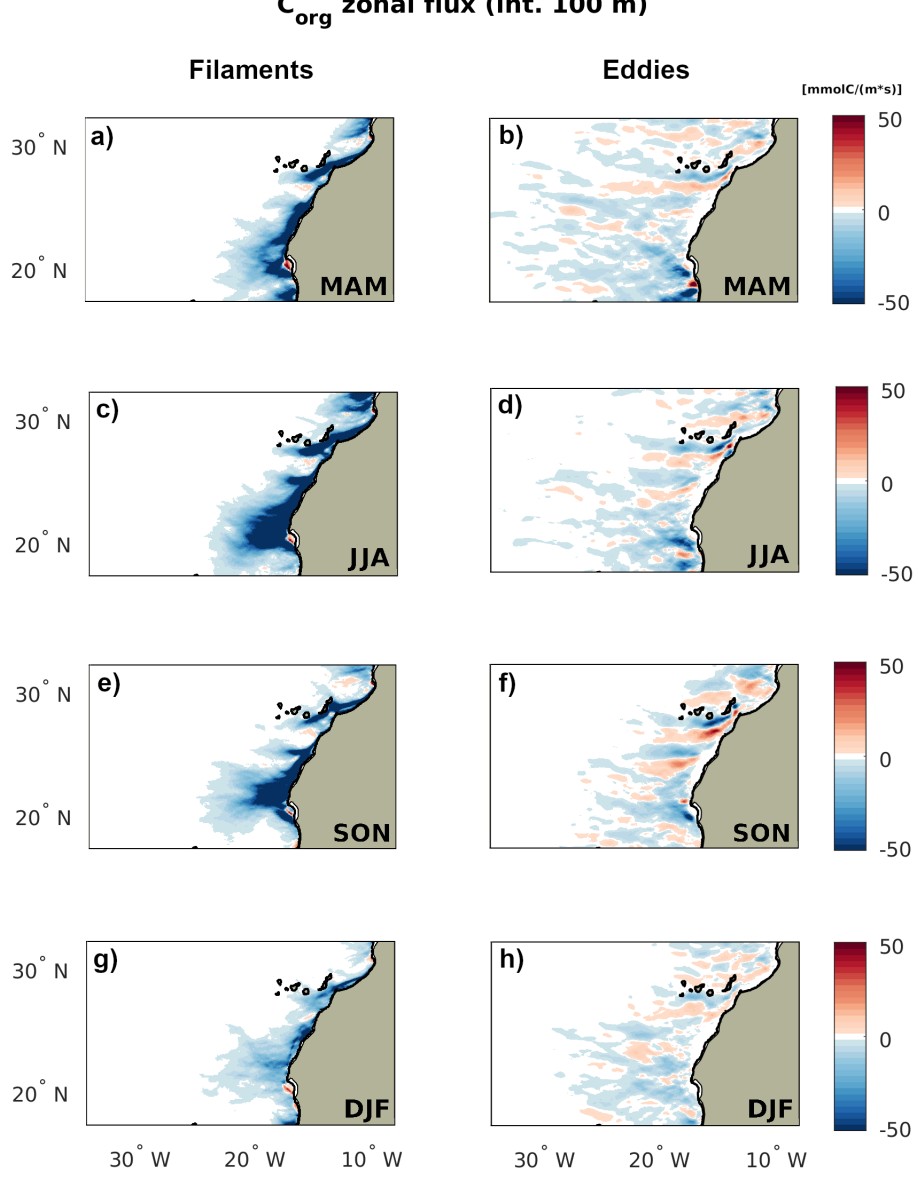

**Figure 3.** Seasonality of the eddy (cyclonic + anticyclonic) and filament offshore transport of $C_{org}$. The first column represents the filament transport. The second column represents the eddy transport. The flux is integrated over the top 100 m depth. Subpanels: (a) and (b) spring (MAM), (c) and (d) summer (JJA), (e) and (f) fall (SON), (g) and (h) winter (DJF).

Hovmöeller diagrams allow us to better assess the propagative nature of the offshore transport and quantify the time it takes for the seasonal $C_{org}$ variations to reach and therefore impact the open waters (Figure 5). As expected, nearshore $C_{org}$
concentrations peak in summer in the NSR and in spring in the CSR, while the widespread spring offshore maximum in $C_{org}$ in

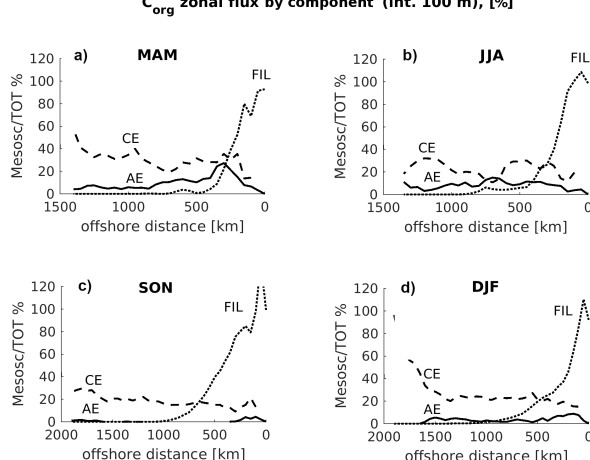

**Figure 4.** Seasonality of eddy and filament contribution to the total offshore transport of $C_{org}$ over CanUS. The plots show the percentage of the total offshore $C_{org}$ flux represented by the filament flux (FIL), the cyclonic eddy flux (CE) and the anticyclonic eddy flux (AE). The mean seasonal eddy and filament components are positive when integrated over the region of study. Percentages go above 100 % when the non-mesoscale flux is negative, bringing the total flux below the sum of the mesoscale flux alone. The flux is integrated throughout the first 100 m depth. Subpanels: a) spring (MAM), b) summer (JJA), c) fall (SON), d) winter (DJF).

the NSR is linked to the North Atlantic bloom dynamics. Both regions show the clear signature of offshore propagating signals in the form of stripes of elevated $C_{org}$ concentrations. In the nearshore, fast signals span roughly the first 250 km offshore in the NSR and the first 500 km offshore in the CSR. Their offshore reach as well as their propagation speed of about 500 km in less than one season reflects the filament dynamics (Lovecchio et al., 2018), and constitute a fast link between the nearshore

temporal variations in biological activity and the adjacent open waters. Farther offshore, slower propagating signals visibly cross the entire domain of interest, spanning about 1000 km in 1 year ( 3 cm s$^{-1}$), in line with the eddy propagation speed at these latitudes (Chelton et al., 2011). These slow propagating signals are likely responsible for a lag in the response of the offshore biological activity to the nearshore seasonal dynamics of upwelling and production.

## 4.3 Quantification of the seasonal variability of the offshore flux by subregion and offshore distance

Our analysis highlights important differences between NSR and CSR in terms of the seasonal and spatial variability of the offshore flux of $C_{org}$, as well as in terms of the magnitude of the contribution of each structure to the total flux (Figure 6). Overall, offshore fluxes in the CSR are higher and have a larger offshore extension than in the NSR.

The NSR shows a pronounced seasonal variability in the fluxes, especially in the nearshore, at the boundary of the coastal region directly influenced by the upwelling. The flux at 100 km offshore peaks in summer, when the upwelling strength
is maximum. In summer, a total of about 160 Gmol C are exported laterally away from the first 100 km from the coast, corresponding to about 1/3 of local nearshore NCP (Figure 7). This maximum in the offshore flux is driven by filaments and eddies, and accompanied by a negative onshore recirculation by the non-filament-non-eddy flux. In both summer and fall, when

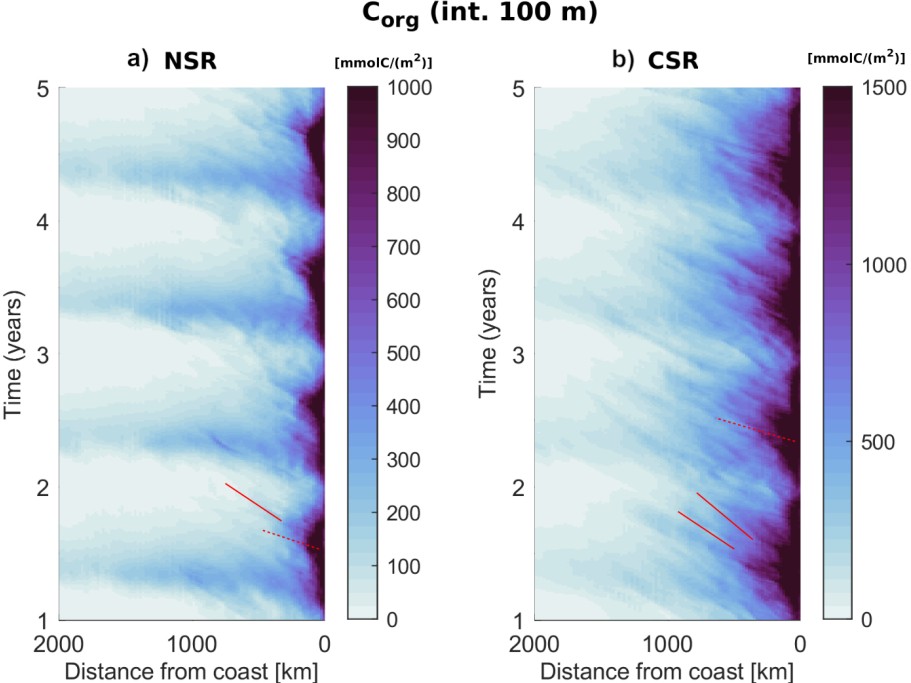

**Figure 5.** Hovmöller diagrams of the $C_{org}$ integrated across the euphotic layer (100 m depth) and averaged across each subregional domain in the first 4 years of analysis data (2-day means). Dotted red lines highlight fast propagating $C_{org}$ signals in the nearshore, while solid red lines highlight slower propagating $C_{org}$ signals offshore.

the NSR upwelling is active, the offshore flux decreases quickly moving away from the coast to the 500 km offshore boundary. At 500 km offshore, in fact, the filament contribution is negligible due to the small extension of the northern filaments (< 300

km), and the flux magnitude becomes between 1/4 and 1/5 of the flux at 100 km offshore. At larger distances, the offshore flux decreases to near-zero values, with an inversion of sign at 1500 km from the coast in summer, likely due to the influence of the onshore flowing Azores current at the northernmost boundary (see also Figure 2b). In Winter and spring, when the upwelling is not active, the offshore flux is smaller near the coast, but it is characterized by a smoother offshore gradient. This is likely due to the time it takes for the nearshore seasonal peak in the $C_{org}$ flux to propagate offshore beyond the range of distances covered

by the filaments. Moreover, in spring $C_{org}$ concentrations are higher also in the open NSR thanks to the north Atlantic bloom.

     The offshore flux of $C_{org}$ in the CSR is characterized by moderate seasonal variability and intense fluxes. The offshore decrease in the intensity of the flux is less dramatic than in the NSR. The flux at 100 km peaks with nearly equal values in spring and summer, but it is still intense in fall and, to a lower extent, in winter. In spring, summer and fall, this flux exceeds 70 % of the NCP produced in the first 100 km from the coast, therefore constituting an extremely large lateral redistribution of

organic material produced at the coast. Filaments are responsible for the large majority of the offshore flux at 100 km, while eddies often exceed the rather small contribution of the non-eddy-non-filament flux. Differently from what we find in the NSR, AE have a moderate but still relevant contribution to the total flux especially in spring, when the upwelling is strongest south




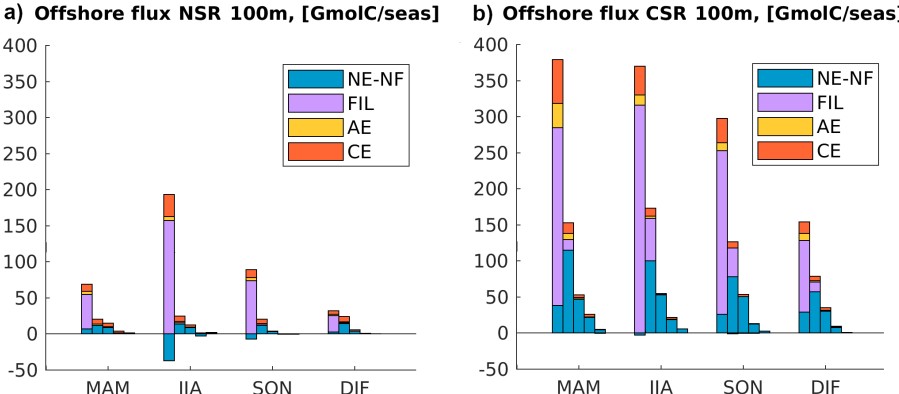

**Figure 6.** Magnitude of the offshore flux of $C_{org}$ integrated in the first 100 m of depth in (a) northern and (b) central CanUS. Each group of bars represents a different season. For each season, each one of the five bars represents the flux through a specific offshore boundary, i.e. an isoline of distance from the coast. From left to right in each group: 100 km offshore, 500 km offshore, 1000 km offshore, 1500 km offshore, 2000 km offshore. Colors in each bar represent the flux contribution by a certain type of structure: non-eddy-non-filament (NE-NF), filaments (FIL), anticyclones (AE), cyclones (CE). Positive and negative contributions to the flux through each boundary are plotted separately in order to make them clearly visible.

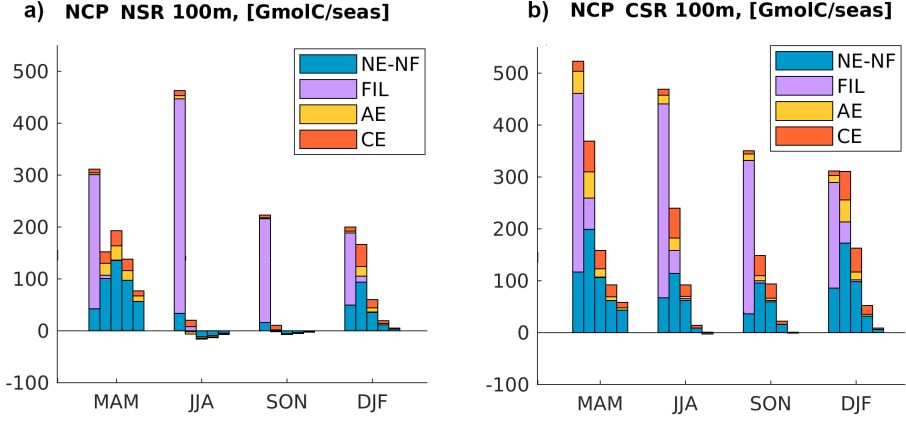

**Figure 7.** NCP in (a) northern and (b) central CanUS integrated in the first 100 m depth and in each offshore region by season. Each group of bars represents a different season. For each season, each one of the five bars represents the difference between the input and the output of offshore flux within two specific offshore boundaries (divergence) or the NCP integrated in the area contained within those two boundaries, i.e. isolines of distance from the coast. From left to right in each group: 0 to 100 km offshore, 100 km to 500 km offshore, 500 km to 1000 km offshore, 1000 km to 1500 km offshore, 1500 km to 2000 km offshore. Colors in each bar represent the contribution by a certain type of structure: non-eddy-non-filament (NE-NF), filaments (FIL), anticyclones (AE), cyclones (CE). Positive and negative contributions to the flux through each boundary are plotted separately in order to make them clearly visible.





of Cape Blanc. This can be explained by the fact that some southern eddies may be shed by the northward-flowing MC, which

likely generates $C_{org}$ rich AE. At 500 km offshore, the filament flux is still important, differently from what we found in the

NSR. This is due to the large extension of the Cape Blanc filament, which is located just in the middle of the CSR. At distances

larger than 500 km from the coast, the non-eddy-non-filament flow, likely in the form of the Cape Verde frontal circulation, is

responsible for the majority of the offshore flux.

### 4.4 Offshore flux divergence variability and its impact on the open waters

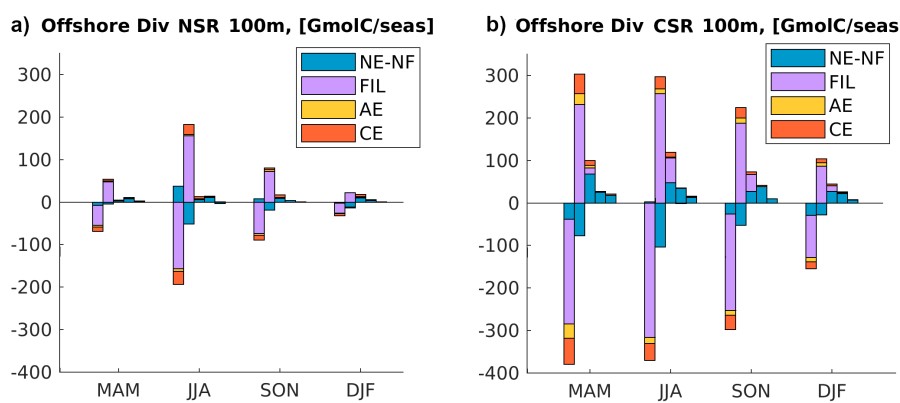

**Figure 8.** Divergence of the offshore flux of $C_{org}$ integrated in the first 100 m of depth in (a) northern and (b) central CanUS. Each group of bars represents a different season. For each season, each one of the five bars represents the difference between the input and the output of offshore flux within two specific offshore boundaries (divergence), i.e. isolines of distance from the coast. From left to right in each group: 0 to 100 km offshore, 100 km to 500 km offshore, 500 km to 1000 km offshore, 1000 km to 1500 km offshore, 1500 km to 2000 km offshore. Colors in each bar represent the contribution by a certain type of structure: non-eddy-non-filament (NE-NF), filaments (FIL), anticyclones (AE), cyclones (CE). Positive and negative contributions to the flux through each boundary are plotted separately in order to make them clearly visible.

The divergence (div) of the offshore flux quantifies the net amount of $C_{org}$ that is added (div > 0) or removed (div < 0)

by the offshore flux from each offshore domain. In steady-state, this divergence is balanced by the local balance between

production and remineralization (net community production) or vertical exchange fluxes. Our results (Figure 8) show that,

in both subregions and in every season, div < 0 in the 0 - 100 km offshore domain, the most productive coastal band. This

means that the cross-shore flux always removes $C_{org}$ from this nearshore region and transports it offshore towards the open

waters. However, the absolute magnitude of this $C_{org}$ transport out of the 0 - 100 km offshore domain differs between NSR and

CSR, and changes substantially between seasons in the NSR. Further away from the 100 km offshore boundary, subregional

differences in the lateral relocation of $C_{org}$ are even more striking both in sign and in seasonality.

In line with all the previous findings, the NSR shows a sharp peak in the summer fluxes. In this season, the amount of $C_{org}$

removed from the 0 - 100 km offshore domain by the offshore flux (div < 0) is at least twice as large than that in all the other

seasons. Compared to the winter, when its value is minimum, the divergence is four times larger. At 100 km off the coast, the



offshore flux relocates about 1/3 of the entire nearshore NCP towards the open waters in both summer and fall. The amount of $C_{org}$ deposited in the range of 100 - 500 km (div > 0) by the offshore flux also peaks in summer and, to a lower extent, in fall, confirming that this range of distances is directly impacted by the seasonality of the nearshore fluxes. This local enhancement of $C_{org}$ availability is especially large if compared to the very low values of summer and fall NCP in the same offshore domain. An analysis of the alongshore fluxes of $C_{org}$ (see Appendix, Figure B2) show a large southward flow of organic material in

the range of 100 km - 500 km offshore as the southern boundary of the NSR in summer, which indicates that in this range of distances part of the $C_{org}$ is further displaced towards the CSR before sinking. This alongshore displacement is connected to the intense southward flow of the Canary Current, which is maximum at distances larger than 100 km from the coast (Pelegrí and Peña-Izquierdo, 2015). Offshore of 500 km from the coast, the NSR shows a weakly negative summer and fall mean of NCP in the euphotic layer, meaning that the surface waters must rely on the lateral input of $C_{org}$ in order to sustain the excess

heterotrophic activity (see also Appendix, Figure B3). This input of $C_{org}$ offshore results from a combination of several lateral fluxes: the delayed signal of the seasonal nearshore flux reaching the open waters in nearly a year time (Figure 5a), the spring relocation of the $C_{org}$ produced locally in the open waters by the North Atlantic spring bloom and nevertheless transported further offshore by the currents (Figure 6a) and, potentially, any additional alongshore input of $C_{org}$ connected to the gyre circulation. Further, an analysis of the model generated eddy tracks (not shown) and meridional fluxes reveals that a significant

northwards $C_{org}$ influx happens at the southern NSR boundary. Here the long and farthermost tail of the giant Cape Blanc filament oscillates and sheds $C_{org}$ rich eddies, which drift northwards while moving away from the coast. This is also visible in the Hovmöller diagram of $C_{org}$ in the NSR (Figure 5a) in the form of high concentration signals that seemingly form at about 1000 km from the coast, and continue to propagate offshore. The picture is extremely different in spring and winter, when the deepening of the North Atlantic mixed layer enhances productivity offshore, therefore turning NCP positive in the open waters.

This increased local productivity of the open waters, combined with a weakening of the offshore flux, reduces the impact of the lateral $C_{org}$ transport on the NSR offshore ecosystem between December and May.

In the CSR, both the seasonal variations and the offshore gradient in the horizontal divergence of the zonal fluxes are less pronounced compared to the NSR. Spring and summer show remarkably similar patterns of the divergence, with the offshore flux transporting away from the first 100 km from the coast at least 60 % of the nearshore production. In fall, the offshore

flux at 100 km offshore is as large as about 90 % of the coastal production at these latitudes. This extremely large offshore relocation of coastal-derived $C_{org}$ is possible also thanks to the lateral convergence of organic material into the CSR from both northern and southern latitudes of the CanUS: the southward flowing CC and CUC and the northward flowing MC contribute to displacing and accumulating $C_{org}$ towards Cape Blanc (Lovecchio et al., 2017). The alongshore influx of $C_{org}$ into the CSR was demonstrated to represent on average about 30 % of the annual mean coastal production. In this sense, the CSR exports

offshore not only its local production but also a fraction of the $C_{org}$ produced in the adjacent CanUS subregions. Most of this $C_{org}$ is added to the 100 - 500 km offhore domain. However, differently from what we found for the northern latitudes, the divergence decreases smoothly offshore, possibly due to the combined influence of the Cape Blanc filament and of the Cape Verde front, which extend far into the open Atlantic. For this reason, the horizontal divergence approaches and sometimes exceeds the magnitude of the local NCP also at the farthest offshore boundary of the CSR. Despite this lateral influx of $C_{org}$,





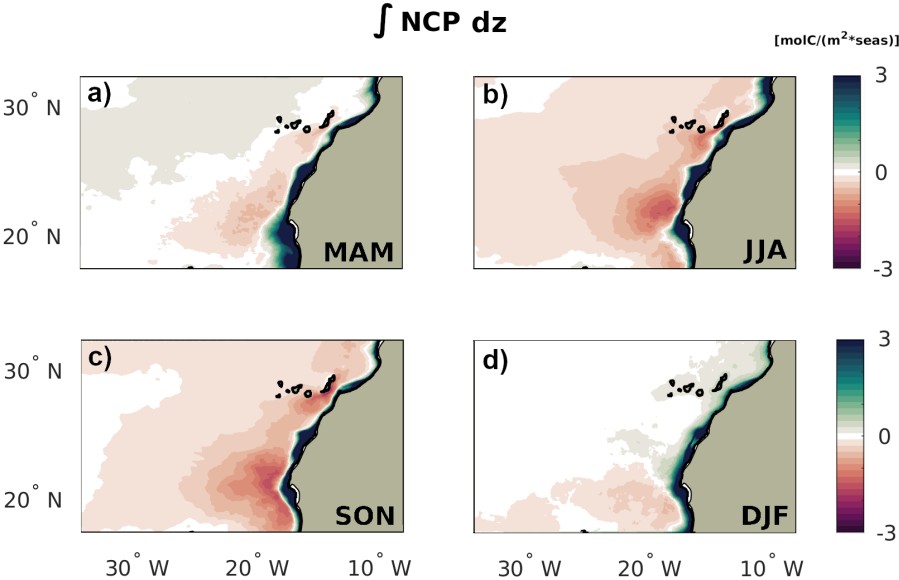

**Figure 9.** NCP by season integrated across the entire water column. Positive regions are net sources of organic carbon, permitting these regions to export $C_{org}$. Negative regions remineralize more organic carbon than what they produce throughout the water column, requiring a net transport convergence of $C_{org}$.

we do not find significant levels of net heterotrophy in the near-surface euphotic layer at these latitudes (see also Figure B3). This may be due to the positive signature of the wind stress curl, especially south of the Cape Verde front, associated with the anticyclonic tropical circulation. This positive wind stress curl contributes to sustaining high levels of offshore primary production.

The spatiotemporal variability of the lateral fluxes of $C_{org}$ in the nearshore and the associated delay in the relocation of the
coastal $C_{org}$ at large offshore distances determines the seasonal and zonal variations in net community production integrated over the entire water-column of the adjacent offshore waters (Figure 9). While the coastal upwelling band has a positive water-column NCP at all latitudes, i.e., represents a net source of organic carbon, beyond 100 km from the coast, the picture becomes more complex. In the NSR, the magnitude of the seasonal fluxes is reflected in the pattern of negative water-column NCP, i.e., water-column net heterotrophy. During summer and fall, this offshore heterotrophy is widespread and far reaching, even
though values remain low beyond 500 km from the coast. This reflects the limited offshore reach of the quick $C_{org}$ transport by upwelling filaments at these latitudes. In spring, the North Atlantic spring bloom induces high levels of offshore production in the open waters of the northern CanUS. Therefore, the offshore water-column heterotrophy of the NSR is confined strictly to the first 500 km from the coast. In winter, the weakening of the offshore transport combined to a deepening of the mixed layer depth and therefore high euphotic layer production offshore (see also Figure B3) result in a neutral water-column NCP.
In the CSR, the intense $C_{org}$ offshore fluxes are reflected in a persistent water-column heterotrophy offshore, even though there exists substantial seasonal variations in the intensity and spatial distribution of the water-column NCP. Despite the intense





lateral fluxes in spring, the excess consumption of $C_{org}$ by remineralization is maximum in summer and fall in the offshore regions, which are reached by the coastal $C_{org}$ signal with a delay of at least one season. Moreover, while offshore fluxes are large in spring, so is also offshore production, especially away from the first 500 km offshore (Figure 7). Since the ratio

between the organic carbon released by the lateral fluxes offshore and local offshore production determines the net heterotrophic activity, spring only shows significant negative water-column NCP between roughly 100 km and 700 km offshore, where the intense filament transport is still relevant. In winter, the offshore water-column of the CSR is weakly heterotrophic, showing that at these latitudes the deepening of the mixed layer depth (and therefore increase in surface production) is not enough to compensate the intense and far-reaching offshore transport of $C_{org}$.

**4.5 Physical and biological drivers of the nearshore flux variability**

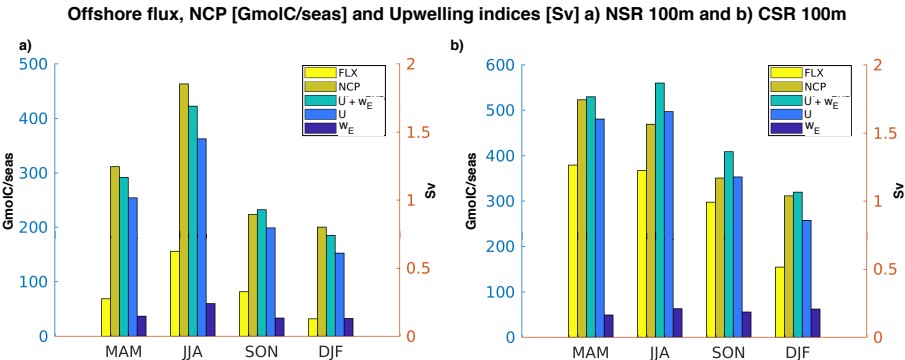

**Figure 10.** $C_{org}$ offshore Fluxes (FLX), NCP fluxes (NCO), upwelling transport ($U + w_E$), coastal upwelling ($U$) and Ekman Pumping ($w_E$) in (a) northern and (b) central CanUS by season. Fluxes are integrated along the coast and in the first 100 m and they represent the flux through the 100km offshore boundary. $U$ is integrated along the coast, $w_E$ is integrated along the the latitude and in the 100km offshore (see section 2.3 for upwelling estimations definition). Each group of bars represents a different season.

The efficiency of the offshore transport of organic carbon is strongly dependent on the coastal production of $C_{org}$, which is sustained by the local upwelling of nutrient-rich deep water via Ekman Transport and Ekman Pumping. Overall, our results show that over the entire CanUS the $C_{org}$ offshore flux at 100km from the coast varies seasonally and in phase with both NCP and Total upwelling (sum of coastal upwelling, $U$ and Ekman pumping, $w_E$) (Figure 10).

Comparing the two sub-regions, the ratio between offshore transport at 100 km offshore and NCP in the first 100 km from the coast is higher in the CSR in all seasons, due to the presence of the giant Cape Blanc filament (Figure 3, see also Appendix: Figure B6). The seasonal variation of the upwelling processes results more pronounced along the NRS, similarly to the fluxes (e.g. offshore flux and NCP flux, see section 4.3). Nevertheless, the yearly mean upwelled volume in the two regions is comparable: the NSR supplies 1.2 Sverdrup (Sv) of upwelled water at surface while the CSR supplies 1.5 Sv.

In the NSR the total upwelling ($U + w_E$) peaks in summer and is still intense in spring, similarly to the other biological and lateral transport fluxes in the subregion (Figure 10a). Upwelling brings up to the surface an estimated volume of 1.8 Sv and



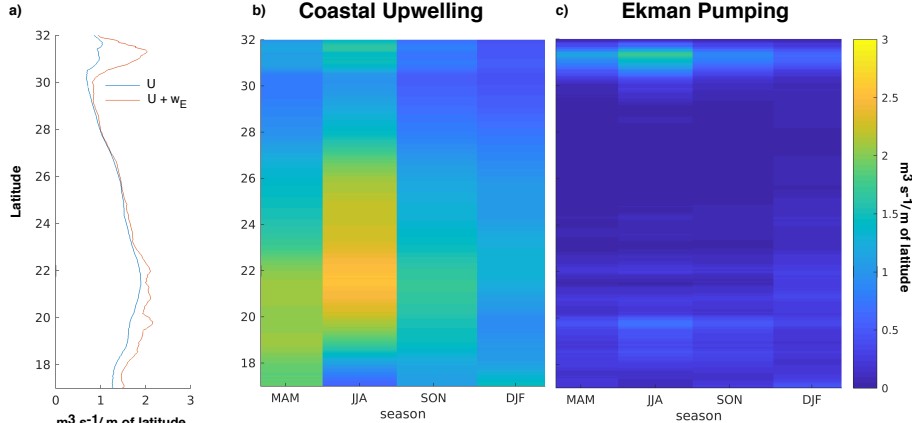

**Figure 11.** (a) Latitudinal distribution of annual mean total upwelling (red line) and Ekman transport (blue line). The difference between the two represents Ekman pumping. Units are in m3 s-1 per latitude meter. Hovmöller diagram (season-latitude diagram) of Coastal upwelling transport (b) and Ekman pumping transport (c) in the first 100km offshore. See section 2.3 for for upwelling estimations definition.

1.2 Sv in summer and spring, respectively. Weak wind stress and wind stress curl at the coast during winter (see Appendix: Figure B4c and Figure B5c) induce a weaker upwelling of about 0.8 Sv, while NCP and spacially-integrated offshore fluxes show their minimum values of 200 and 32 GMolC/seas. Autumn is characterized by a more efficient offshore flux of $C_{org}$ away

from the coast compared to spring, despite the weaker total upwelling (1.0 Sv) and NCP. This can be explained by the fact that the filaments during fall are more intense and persistent along the coast with respect to spring, especially near Cape Juby (see Appendix: Figure B6).

The fluxes and the upwelling processes in the CSR are characterized by moderate seasonal variability (Figure 10b). The total upwelling peaks in summer (1.8Sv) and it shows nearly equal values in spring (1.7Sv). It is still intense in fall (1.3Sv)

and to a lower extent, in winter (1Sv). Even thought upwelling maxima is in summer, the NCP flux, and in turn the $C_{org}$ flux, are slightly more intense in spring. This could be explained, as presented before, by the $C_{org}$ sinks in summer, characterized by nearly neutral or slightly heterotrophic water-column nearby the coast (e.g. north of Cape Blanc, Figure B3b). In contrast to what we find for NSR the two upwelling processes show different seasonality. Either the coastal upwelling ($U$) and the Ekman pumping ($w_E$) peak during summer, but the latter is still strong in winter (0.22Sv), without showing a maximum in spring. The

reason for this discrepancy and for the moderate seasonal variability of the upwelling is embedded in the definition of the CSR sub-region. The CSR expands from 17 °N to 22 °N and it includes a large portion of ocean around Cape Blanc (21 °N). This choice was made in order to properly evaluate the offshore transport by the giant Cape Blanc filament, which accounts for a large amount of lateral export of particulate matter (Gabric et al., 1993). Nevertheless, Cape Blanc bounds two zonal bands of the CanUS which are characterized by different seasonality of alongshore winds and, in turn, of the upwelling (Arístegui et al.,

2009). Hovmöller diagrams of costal upwelling (Figure 11b) and Ekman pumping (Figure 11c) reveal pronounced upwelling north of Cape Blanc in summer and spring. By contrast, south of Cape Blanc, the upwelling processes are more robust in





winter and spring. The combination of these two different upwelling peaks results in a mixed seasonal character for the CSR. In particular, the seasonality of the coastal upwelling is dominated by the strong alongshore winds (Figure B4) during summer north of Cape Blanc, while the Ekman pumping seasonality, which depends on the gradient of the winds, is affected by the

winter to spring upwelling variability south of Cape Blanc.

Howmöller diagrams and annual mean upwelling estimations along latitude (Figure 11a) allow the comparison between the two different upwelling processes along the coast. In particular, coastal upwelling is the dominant mechanism along the latitudes, accounting up to 80% of the total upwelling. A local exception is found south of Cape Ghir (31°N) where Ekman pumping plays a crucial role likely due to the coastline geometry, the bay south of Cape Ghir prevents the direct effect of the

along shore winds (see Appendix Figure B4).

## 5  Discussion

### 5.1  Comparison with previous studies

Seasonality in the CanUS varies substantially with latitude, allowing to identify a set of subregions with distinct peaks in upwelling and production and distinct levels of mesoscale activity (Arístegui et al., 2009; Pelegrí and Peña-Izquierdo, 2015). Our

model is agrees with previous studies which distinguished between a northern region of summer upwelling intensification, a central zone of semi-permanent upwelling, and the southern latitudes characterized by a late winter to spring peak of upwelling. In the Moroccan sector (21°N-34°N), permanent equatorward winds and positive wind stress curl sustain Ekman Transport and Ekman pumping with a summer intensification. An inversion of the alongshore flow in the proximity of the Canary Islands (Lanzarote passage) is visible from Figure B2 and was previously described (Machín et al., 2010; Mason et al., 2011). In

the northern part of the Mauritanian-Senegalese sector (17°N-21°N), the meridional shift of the trade winds causes seasonal upwelling with a late winter to spring peak (Lathuilière et al., 2008; HAGEN, 2001). The magnitude of the coastal upwelling transport and its latitudinal and seasonal variability (Figure 11) are consistent with previous studies, such as Benazzouz et al. (2014) and Desbiolles et al. (2014). Seasonality in the offshore flux gets smoother at lower latitudes, where a combination of both mean and mesoscale flow extend the impact of the transport up to the farthest offshore boundary of our analysis domain.

Previous literature on the offshore transport of $C_{org}$ in the CanUS based on in-situ observations provided some estimates of the offshore transport in the CanUS, with most of the studies being focused on the NSR range of latitudes. Many of these studies used a box-model approach and refer to a specific time and sub-sector of the CanUS coastline, therefore integrating the lateral $C_{org}$ flux over domains of different meridional extension. In order to provide a fair comparison, we discuss here our results and previous results in terms of fluxes per degree of coastline ($deg^{-1}$). Both Garcìa-Muñoz et al. (2004) and Santana-Falcón

et al. (2016) provided estimates of the $C_{org}$ flux for sub-sectors of the NSR in August 1999 and August 2009, respectively. Our summer $C_{org}$ flux in the NSR at 100 km offshore corresponds to an average 32.9 kgC $s^{-1}$ $deg^{-1}$, comparing to 75.2 kgC $s^{-1}$ $deg^{-1}$ (Garcìa-Muñoz et al., 2004) and 20.9 kgC $s^{-1}$ $deg^{-1}$ (Santana-Falcón et al., 2016). Referring to September-October 1997, García-Muñoz et al. (2005) calculated 7.0 kgC $s^{-1}$ $deg^{-1}$; in comparison, we obtain a mean fall flux of 16.4 kgC $s^{-1}$





deg$^{-1}$ at 100 km offshore. Considering the different averaging times, lateral extension of the sampled area and the substantial
small scale variability affecting the nearshore CanUS, our results fall close to what was found by in-situ observations.

Messié and Chavez (2015) studied the seasonal variability of net primary production in the EBUS and they identified the
wind driven nutrient supply (e.g. calculated using upwelling indices) as the major driver of the seasonal variability of net
primary production. In agreement with the findings by Messié and Chavez (2015), our results indicate overall a strong positive
correlation between the magnitude of the upwelled water volume and the strength of net community production in both sub-
regions. Likewise, high NCP is strongly correlated with high offshore fluxes of C$_{org}$. Thus, upwelling processes are the primary
drivers of C$_{org}$ variability. In agreement with Messié et al. (2009), coastal upwelling results the dominant process in CanCS
along all the latitude, except north of Cape Sim (31°N) where Ekman Pumping gets stronger. The relative importance of the two
upwelling components, is a function of coastal topography and therefore there is strong latitudinal variability. Ekman transport
is higher next to capes (31°N;21°N) or just downstream due to alongshore wind acceleration, whereas Ekman pumping is
stronger in the lee of the capes (29°N) where winds weaken while remaining high offshore, increasing the cyclonic curl
(Bakun and Nelson, 1991; Pickett and Paduan, 2003; Koračin et al., 2004).

Our seasonal analysis of the impact of the offshore transport of C$_{org}$ onto the open waters of the NASG provides further
insight about the net metabolic state of the near-surface open waters in low productive areas (Williams et al., 2013; Duarte
et al., 2013; Ducklow and Doney, 2013). Even though in the annual mean the CanUS euphotic layer was proven to be net
autotrophic (Lovecchio et al., 2017), our new results show that in summer and fall the offshore waters of the NSR present
weak levels of net heterotrophy in the first 100 m depth (Figure 7 and Figure B3). This excess heterotrophy is fueled by the
lateral redistribution of C$_{org}$ in the offshore and alongshore directions combined with the shoaling of the mixed layer depth in
the warm seasons, and is maximum in specific hotspots located in the offshore region adjacent to the coastal upwelling band
(roughly between 100 and 200 km offshore). When compared to the net autotrophic activity of the spring-winter season and to
the intense autotrophy of the upwelling area, these spots of near-surface net heterotrophy are weak. However, they may indicate
that measurements made in the summer and fall in the offshore NSR could highlight weak levels of excess remineralization,
i.e., negative NCP.

## 5.2 Model limitations

Our climatological run was forced with monthly mean winds derived from ERA-Interim (Dee et al., 2011), which, at the
time of testing, provided an overall better performance of the model compared to other QuickSCAT products (Risien and
Chelton, 2008). Despite the well known biases of Era-Interim product at coast (Bonino et al., 2019b; Taboada et al., 2019),
our upwelling estimations are in agreement with Messié et al. (2009) estimations, which compute vertical transports from
QuikSCAT winds. This is likely due to the fact that we computed the coastal upwelling transport from the winds interpolated
on the Atlantic telescopic grid and we considered the mean value of the alongshore winds within 100km from the coast in
order to be consistent with the computation of the fluxes, which are estimated at the 100km offshore boundary. Nevertheless,
a higher resolution global data set of winds would allow for a more accurate assessment of the relative proportion of transport
and pumping over CanCS.





The NPZD model employed in the present study does not include a dissolved organic carbon (DOC) pool, meaning that we may be underestimating the lateral transport of organic material in the euphotic layer given that both DOC is subject to little

vertical export. This may be especially relevant in small scale structures such as filaments, which are known to contain high percentages of DOC (Santana-Falcón et al., 2016; García-Muñoz et al., 2005) and which we demonstrate to have a key role in driving the seasonal variations of the offshore transport near the coast. In terms of impact on the offshore biological activity, however, only a small fraction of DOC (the labile fraction) is expected to play a significant role in fueling heterotrophic activity (Hansell et al., 2009). Moreover, the slow sinking rate of our small particle pool (1 m s$^{-1}$) means that modeled small POC

behaves quite similarly to a suspended pool, therefore allowing significant lateral transport and potentially compensating the lack of a DOC pool. We refer the reader to (Lovecchio et al., 2017, 2018) for a discussion of the strengths and weaknesses regarding the analysis of the lateral fluxes of $C_{org}$ in the absence of a modeled DOC pool.

## 6 Conclusions

The CanUS is characterized by the most intense seasonal fluctuations in physical and biogeochemical fluxes among all EBUS

(Chavez and Messié, 2009) and by an offshore transport of $C_{org}$ that determines $C_{org}$ availability in the adjacent open waters. Upwelling processes, driven by the surface alongshore winds, play a crucial role in determining the seasonal variations and latitudinal changes of the nearshore $C_{org}$ fluxes.

Our results show that, in the CanUS, seasonal fluctuations of the $C_{org}$ offshore flux are strongly mediated by small scale upwelling filaments, which contribute the most to the total flux at 100 km from the coast in each season and at all latitudes. A

combination of eddy and non-mesoscale lateral fluxes extends the $C_{org}$ transport farther away from the coast, with maximum offshore extension in spring and summer. Additional seasonal variability in the alongshore displacement of $C_{org}$ depicts a dynamic and fully 3D biological pump.

Our results also highlight that temporal variations in nearshore processes such as upwelling, production and coastal circulation determine analogous temporal variations in the magnitude and spatial extension of the $C_{org}$ offshore flux, which is

an essential component of the coastal-open ocean biological coupling. This contributes to modulate the seasonal changes in the trophic state of the open waters at several hundreds of kilometers off the north-western African coast, albeit with some temporal delay. This delay is smaller than a season only in the nearshore range covered by the intense filament transport. As the dominant scales of temporal variability differ for different EBUS (Chavez and Messié, 2009; Frischknecht et al.), further studies are needed to investigate the repercussions for the open ocean biological activity in regions others than the CanUS.

*Data availability.*   Model output is available upon request. Please contact the authors Giulia Bonino (giulia.bonino@cmcc.it) or Elisa Lovecchio (elisa.lovecchio@noc.ac.uk) in that matter. The data have been registered at the ETH-library archive and are available at: https://www.research-collection.ethz.ch/handle/20.500.11850/278536 (last access: 23 October 2020)





## Appendix A: Acronyms

- AC = Azores Current

- AE = anticyclones

- AMO = Altantic Multidecadal Oscillation

- CanUS = The Canary Upwelling System

- CC = Canary Current

- CE = cyclones

- $C_{org}$ = Organic Carbon

- CSR = Central SubRegion of the CanUS

- CUC = Canary Upwelling Current

- CVF = Cape Verde Front

- DJF = December January February (winter)

- EBUS = Eastern Boundary Upwelling Systems

- ENSO = El Niño Southern Oscillation

- FIL = filaments

- JJA = June July August (summer)

- MC = Mauritanitan Current

- MAM = March April May (spring)

- NASG = North Atlantic Subtropical Gyre

- NCP = net community production

- NF-NE = non-filament-non-eddy

- NPP = Net Primary Production

- NPZD = Nutrient Phytoplankton Zooplankton Detritus ecosystem model

- NSR = Northern SubRegion of the CanUS





- ROMS = Regional Ocean Modeling System

- SON = September October November (fall)

- SSH = Sea Surface Height

- SST = Sea Surface Temperature

- $U$ = coastal Upwelling Index

- $w_E$ = Ekman Pumping

**Appendix B: Supplementary Figures**

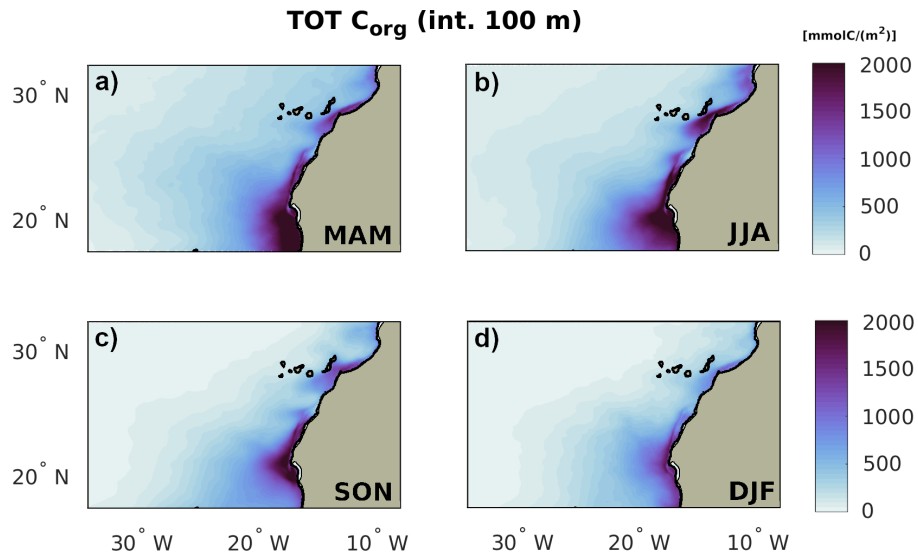

**Figure B1.** Total $C_{org}$ by season in the euphotic layer (100 m depth).

*Author contributions.* GB, EL, DI and NG conceived the study. EL and MM set up the experiment and improved the model. GB and EL
performed the analysis, interpreted the results and wrote the paper. All authors contributed to improving the paper.

*Competing interests.* The authors declare that they have no competing interests.



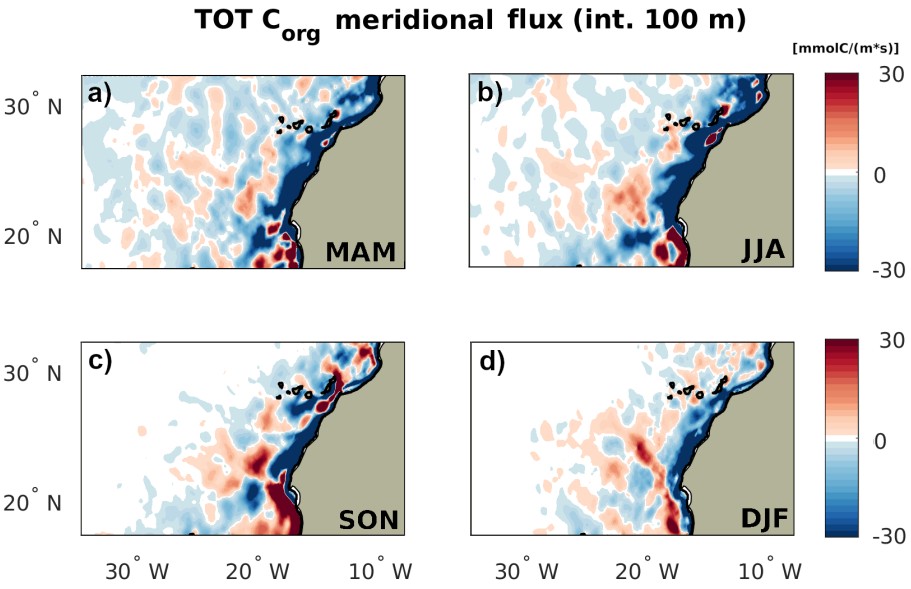

**Figure B2.** Seasonality of the total meridional transport of C_org by season, positive meaning northward. The flux is integrated throughout the first 100 m depth.

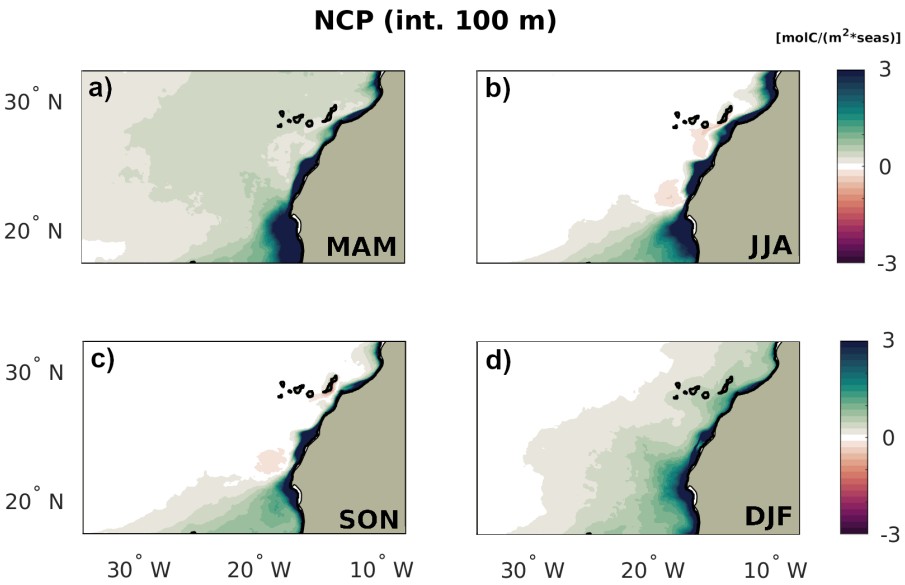

**Figure B3.** Total NCP by season integrated across the euphotic layer (100 m depth).

*Acknowledgements.* This research was financially supported by the Euro-Mediterranean Center on Climate Change (CMCC, Bologna, Italy) and by the Swiss National Science Foundation (Project CALNEX, grant No.149384). The model simulations were performed at the HPC





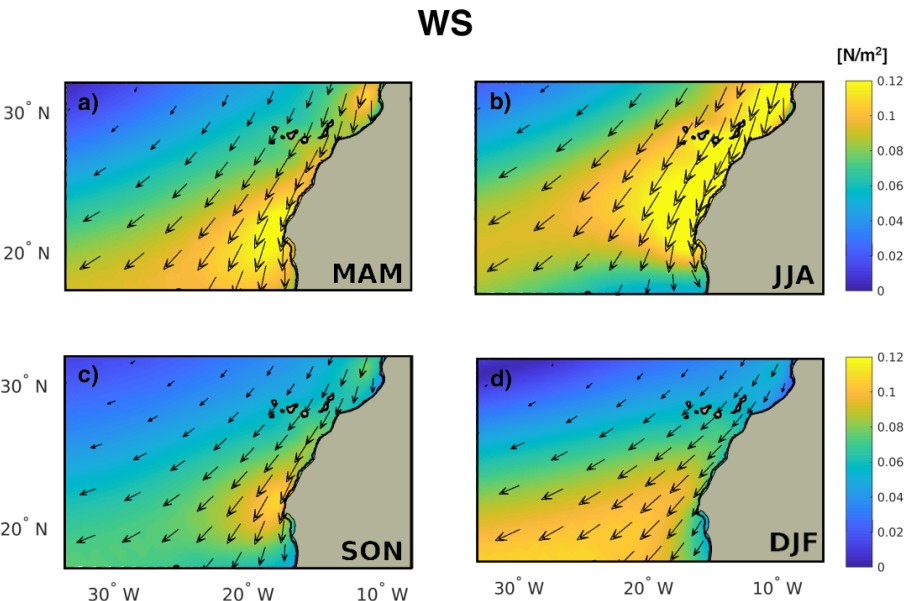

**Figure B4.** Seasonality of the Wind stress.

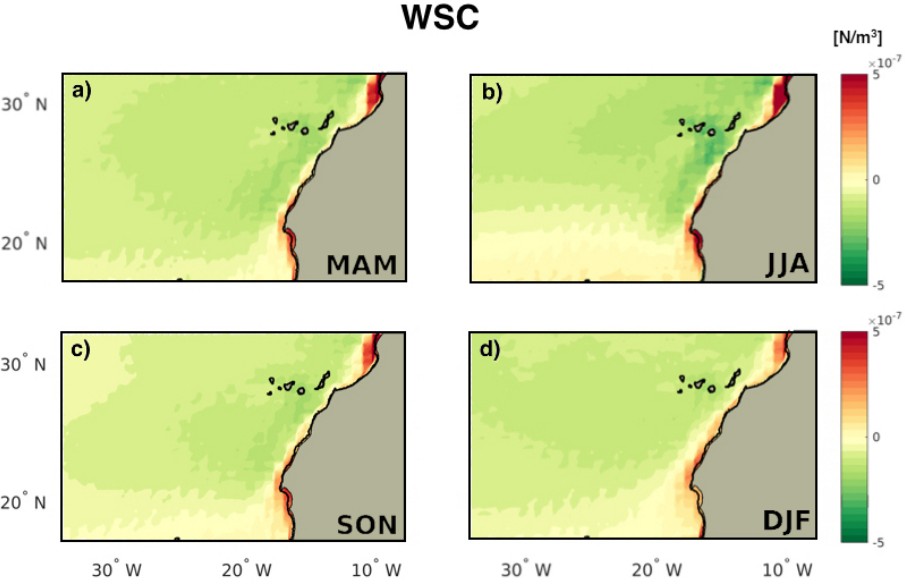

**Figure B5.** Seasonality of Wind stress curl.

cluster of ETH Zürich, Euler, which is located in the Swiss Supercomputing Center (CSCS) in Lugano and operated by ETH ITS Scientific
IT Services in Zurich.



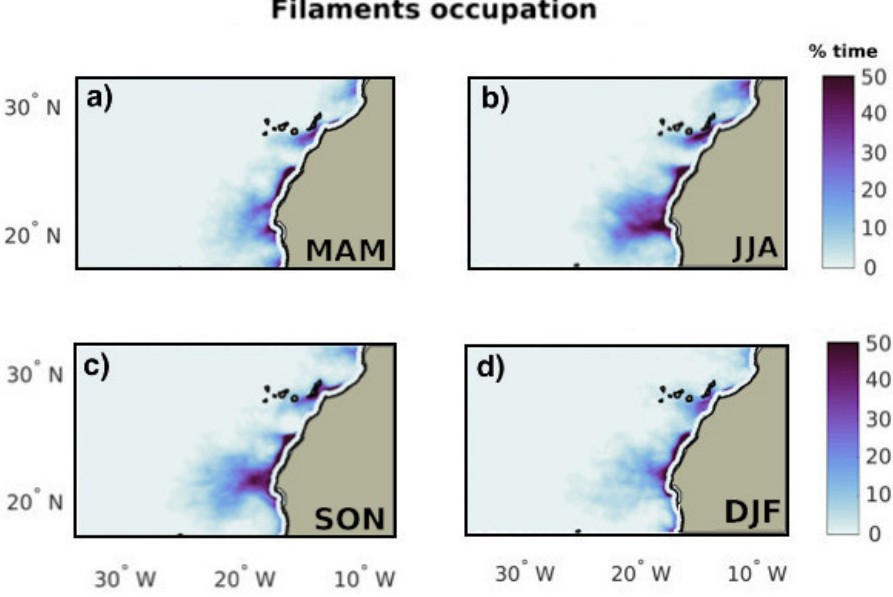

**Figure B6.** Filament occupation by season. The first 50 km from the north-western African coast have been shaded, as by definition filaments always cover this range of distances.

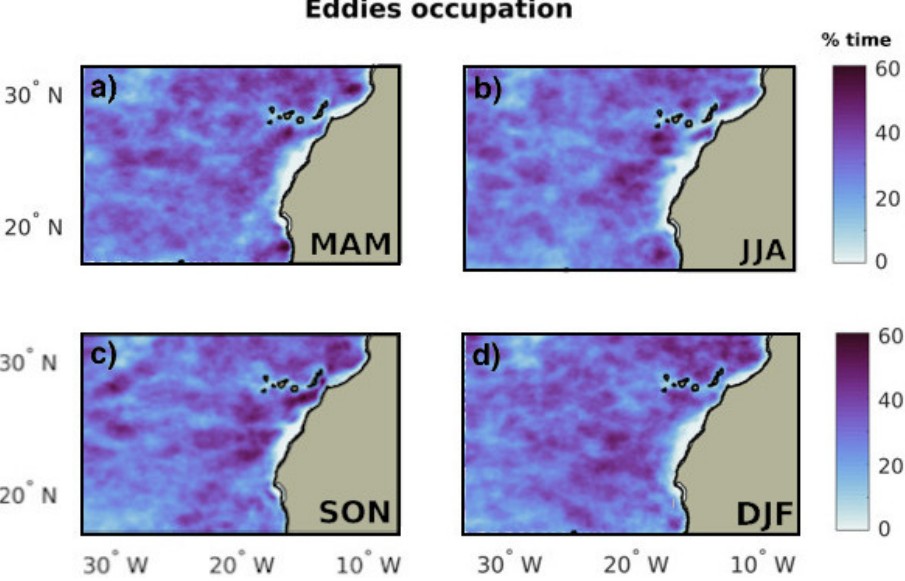

**Figure B7.** Eddies occupation by season.



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
