# Peer review of "Drivers and impact of the seasonal variability of the organic carbon offshore transport in the Canary Upwelling System"

_Biogeosciences, 2020_

## Referee Comment (RC1) · Anonymous Referee #1 · 6 Jan 2021

GENERAL STATEMENT

This paper presents a comprehensive modeling study to test the seasonal variability of the offshore transport of particulate organic carbon (POC) in the Canary Current EBUE from 17° to 32°N. It is based on a coupled hydrodynamic + biogeochemical model that has been developed and previously used by the same research group to respond other pertinent scientific questions about this EBUE (Lovecchio et al. 2017; 2018). The results of the modeling exercise confirm that seasonal differences are significant and dependent on latitude. The authors also explore the physical and biological drivers of this variability. Overall, it is a well-written, suitable contribution to the understanding of

part of the Canary Current EBUE and it is very appropriate for Biogeosciences.

I would not say that the paper is filling the gap of knowledge about the seasonality of the offshore transport of POC (line 5 of the manuscript). It would be more reasonable to say that this modeling exercise have identified that seasonality appears to be significant. This wording would make much happier the many observational oceanographers that would be interested in this work.

It is the author's choice to focus their study on part of the Canary Current EBUE, but it is important to define appropriately this EBUE, which extends from 43°N to 10°N and, therefore, includes also the Atlantic coast of the Iberian Peninsula. Although the model domain does include neither the Iberian coast nor the Strait of Gibraltar, when defining the Canary Current EBUE they should be included (line 37).

Although the authors refer to offshore transport of organic carbon (Corg), they clearly state that only particulate organic carbon (POC) is simulated. It is well-known from previous observational works in the area that the dissolved organic carbon excess in surface coastal waters compared to deep upwelled waters (deltaDOC) represents from 65 to 95% of the total organic carbon export (Corg = deltaDOC + POC) by up-welling filaments (Muñoz-García et al. 2004; 2005; Álvarez-Salgado et al., Limnology & Oceanography, 2007; Santana-Falcón et al., 2016). Note that it is not DOC but delta-DOC, i.e. after subtracting the refractory fraction of DOC. Therefore deltaDOC would represent the labile DOC pool. According to those papers, if the model run here considers only the POC pool, it would underestimate the lateral transport of Corg from 3 to 20 times. But, if the POC export fluxes obtained here are multiplied by 3 to 20, then the Corg export would surpass the NCP, which would be inconsistent. At the end of the manuscript, in the section devoted to model limitations the authors state that the small (slow-sinking) POC pool can potentially compensate the lack of a DOC pool (lines 439-441) and refer to their previous papers (Lovecchio et al., 2017; 2018) for a discussion about the quantification of lateral fluxes of Corg in the absence of the DOC pool. However, I think that this point is sufficiently important to be discussed in this manuscript.

Furthermore, the issue about how good is the small POC pool mimicking the labile DOC pool should be introduced earlier instead of just stating that "no DOC pool is included in the model" (line 100-101). In this regard, the recent work by Santana-Falcon et al. (Progress in Oceanography, 2020) is relevant for this discussion as they used a coupled physical+biogeochemical model that includes a DOC pool.

SPECIFIC COMMENTS

Line 10. I would say "the interaction of the Cape Blanc filament with the Cape Vert front".

Line 30. Longer time scales associated to global warning should also be considered.

Line 34 (and also in Line 458). Frischknecht et al. 20??.

Line 98. Please, indicate integration depth here.

Line 101. See general comments above.

Lines 196. Add "eddy" after "mesoscale".

Line 400. Comparison with the recent modeling study by Santana-Falcón et al. (2020) should be included here.

Lines 433-442. As indicated above, the issue of the contribution of deltaDOC to lateral Corg export is very relevant for the manuscript, should be introduced earlier and discussed in more detail. Note also that although deltaDOC is a small fraction of the total DOC it represents from 3 to 20 times the concentration of POC. Furthermore, it is also likely that the deltaDOC transported offshore accumulates in the adjacent oligotrophic gyre because of nutrient limitation to heterotrophic activity rather than the refractory nature of these materials (e.g. Hansell et al., 2009).

Figures 2 and 3 are somewhat redundant. I would erase Figure 2.

Figure 4. Please, add the non-mesoscale flux contribution too.

Figure 10. Change "NCO" by "NCP" in caption.

---

## Referee Comment (RC2) · Takeyoshi Nagai (Referee) · 11 Feb 2021

This study presents the results from the numerical model for the Canary Upwelling System coupled with a NPZD ecosystem model. The results highlight the rapid offshore transport caused by the filaments with slower but long reaching eddy fluxes. Also, their analyses revealed that the stronger seasonality in the offshore carbon flux in the northern sub region compared to the central sub region. The manuscript is well-written with a number of typos. Therefore, in my opinion, the manuscript should be published after some revisions considering the following points.

The major issues Although the authors mentioned several reasons of the less seasonality in the central sub region, such as the convergence of CC and CUC, and the Cape Blanc filament, the dominant mechanism of this is not clearly written in the conclusion and abstract in the manuscript. I think that the novel finding of this paper is the seasonality variation along the latitude. Therefore, it is important to convey the concise and clear statement for the mechanism, which drives the seasonality variations both in the abstract and the conclusion.

The manuscript includes many sentences without referring any figures, which would make many readers to have difficulty to follow the arguments.

Specific Comments

L12 "every season season" Remove one of "season"s.

L34 "Frischknecht et al." Publication year is missing.

L55 "N)," Remove a parenthesis.

L107 "17âŮęN to 24.5âŮęN]" Remove "]".

L108 "This correspond to" should be "This corresponds to"

L199 "In every season the filament flux is dominated by a negative (and therefore offshore-directed) Corg flux, with only a minor contribution to the onshore Corg recirculation." Why is the filament flux dominated only by offshore flux, despite that the filament has offshoreward and onshoreward currents on its northern and southern parts, respectively?

L247 "In Winter and spring..." Uncapitalize "W".

L306 "offhore" should be "offshore".

L339 "and Total upwelling" "T" should be lowercase.

L340 "the ratio between offshore transport at 100 km offshore and NCP in the first 100 km from the coast is higher in the CSR in all seasons" Is it the ratio of offshore transport

at 100 km offshore to NCP in the first 100 km from the coast, or the other way around?

L342 ". . .results more pronounced along. . ." Pronounced effects?

L342 "NRS" Is it "NSR"?

L345 A comma is needed after "In the NSR".

Fig. 11 caption "m3s-1" Numbers should be uppercase.

L355 "Even thought" should be "Even though". "NCP flux". Is it flux or production rate?

L359 "still strong in winter (0.22 Sv)" It's better to refer Fig. 10.

L380 "model is agrees" should be "model agrees".

L386 "HAGEN, 2001" is all capital. Is it okay?

L406 "CanCS" This appears here at the first time in the paper without any definition.

L414 "in the annual mean the CanUS" Remove "the" in front of CanUS.

L416 "This excess heterotrophy is fueled by the lateral redistribution of Corg in the offshore and alongshore directions combined with the shoaling of the mixed layer depth in the warm seasons, . . ." Before this sentence, the authors mentioned the effects of deepening of the mixed layer, not shoaling. Do you mean that the shoaling of the mixed layer prevents the Corg from being diffused up?

L441 "We refer the reader to (Lovecchio et al., 2017, 2018) for a discussion. . ." should be "We refer the reader to Lovecchio et al., (2017, 2018) for a discussion".

L457 "This delay is smaller than a season only in the nearshore range covered by the intense filament transport." Do you mean that "This delay is typically shorter than one season only in the nearshore range covered by the intense filament transport."?

"

---

## Author Comment (AC1) · 3 Mar 2021

**Answer to Anonymous Referee nr.1**

We thank Anonymous Referee nr.1 for the time spent on reviewing our manuscript and for their thoughtful comments and useful corrections that contributed to improve the quality of our manuscript. We especially appreciated their interest in the discussion of the role of the dissolved organic carbon pool, and we therefore worked on the manuscript in order to provide a more in-depth perspective of this point.

We include our detailed answers to their main and specific comments below. We copied in blue the Referee's comments, and included our answers in black. Changes to the manuscript text are highlighted in italics

**Answers to the main comments**

*MC1) I would not say that the paper is filling the gap of knowledge about the seasonality of the offshore transport of POC (line 5 of the manuscript). It would be more reasonable to say that this modeling exercise have identified that seasonality appears to be significant. This wording would make much happier the many observational oceanographers that would be interested in this work.*

We have rephrased the abstract removing the expression "fill the gap".

*MC2) It is the author's choice to focus their study on part of the Canary Current EBUE, but it is important to define appropriately this EBUE, which extends from 43◦N to 10◦N and, therefore, includes also the Atlantic coast of the Iberian Peninsula. Although the model domain does include neither the Iberian coast nor the Strait of Gibraltar, when defining the Canary Current EBUE they should be included (line 37).*

We acknowledge this comment and agree that often the Iberian Upwelling coastline is included in the definition of the Canary Current System. In order to clarify this point, we have included the following sentence in the introduction:
*"The Iberian upwelling coast is often also included in the definition of the CanUS (Arístegui et al., 2009), although in this manuscript we only focus on the northwestern African sector."*

*MC3) Although the authors refer to offshore transport of organic carbon (Corg), they clearly state that only particulate organic carbon (POC) is simulated. It is well-known from previous observational works in the area that the dissolved organic carbon excess in surface coastal waters compared to deep upwelled waters (deltaDOC) represents from 65 to 95% of the total organic carbon export (Corg = deltaDOC + POC) by upwelling filaments (Muñoz-García et al. 2004; 2005; Álvarez-Salgado et al., Limnology & Oceanography, 2007; Santana-Falcón et al., 2016). Note that it is not DOC but deltaDOC, i.e. after subtracting the refractory fraction of DOC. Therefore deltaDOC would represent the labile DOC pool. According to those papers, if the model run here considers only the POC pool, it would underestimate the lateral transport of Corg from 3 to 20 times. But, if the POC export fluxes obtained here are multiplied by 3 to 20, then the Corg export would surpass the NCP, which would be inconsistent. At the end of the manuscript, in the section devoted to model limitations the authors state that the small (slow-sinking) POC pool can potentially compensate the lack of a DOC pool (lines 439- 441) and refer to their previous papers (Lovecchio et al., 2017; 2018) for a discussion about the quantification of lateral fluxes of Corg in the absence of the DOC pool. However, I think that this point is sufficiently important to be discussed in this manuscript.*

*Furthermore, the issue about how good is the small POC pool mimicking the labile DOC pool should be introduced earlier instead of just stating that "no DOC pool is included in the model" (line 100-101). In this regard, the recent work by Santana-Falcon et al. (Progress in Oceanography, 2020) is relevant for this discussion as they used a coupled physical+biogeochemical model that includes a DOC pool.*

We thank Anonymous Referee nr.1 for this important observation, and agree that the paper can include a more thorough discussion of how the lack of a modeled DOC pool may affect our results.

Since the implied NPZD model is self-consistent and routes the living Corg only into the two sinking pools of non-living Corg accordingly to the model's equations, it is not possible to estimate how the offshore fluxes may change in presence of an additional Corg pool without running in inconsistencies. For this reason, in a previous publication (Lovecchio et al. 2017) based on the same model setup, the authors performed a sensitivity study designed to quantify the impact of a suspended Corg pool on the total organic carbon offshore fluxes and their impact on the open waters. This experiment showed that, even though the magnitude of the lateral flux increased, the divergence of the flux (and therefore the extra organic carbon added offshore for remineralization) remained roughly the same. This suggested that the non-sinking pool had a minor impact on the heterotrophic activity of the deep open waters reached by the offshore flux. We have now summarized the results of this sensitivity study in the manuscript, with the aim of providing a better picture of what may change in the presence of a non-sinking pool of Corg.

Further, we have added some considerations regarding the role of DOC in the filament transport, based on the Santana-Falcon 2016 and 2020 papers that showed the large fraction of DOC contained in such structures. We have therefore added an extra paragraph focusing on this model limitation in the discussion section, and we now explicitly refer to it earlier in the Methods.

In detail, we have made the following changes to the manuscript:

We have added the following sentence to the Methods:

"No dissolved organic pool (DOC) is included in the model; we address this shortcoming in the discussion section."

We have rephrased and extended the dedicated paragraph in the Discussion subsection as follows:

*"Both non sinking and dissolved organic material are expected to play an important role in the lateral Corg fluxes, given that they are subject to little vertical export. In fact, even though only a small fraction of DOC (the labile fraction) is expected to play a significant role in fueling heterotrophic activity, this fraction of DOC can have concentrations that exceed the particulate Corg pool by several times (Hansell et al., 2009). This may be especially relevant in small scale structures such as filaments,which are known to contain very high percentages of DOC (Santana-Falcón et al., 2016, 2020; García-Muñoz et al., 2005). The NPZD model employed in the present study does not include a DOC pool, meaning that we may be underestimating the total lateral transport of organic material in the euphotic layer. Furthermore, since we demonstrate that filaments have a key role in driving the seasonal variations of the offshore transport near the coast, a prevalence of*

*DOC in these type of structures may further exacerbate their role in driving the total Corg offshore flux.*

*In our model, the slow sinking rate of the small particle pool (1 m s−1) means that small POC behaves quite similarly to a suspended pool, therefore allowing significant lateral transport and potentially compensating the lack of a DOC pool. However, in order to better explore the potential impact of this model limitation, Lovecchio et al. (2017) presented the results of a key sensitivity experiment in which the small detritus sinking speed was set to zero, while strongly limiting its coagulation rate. The results of this experiment showed that even though the absolute value of the offshore flux of Corg increased in presence of a non-sinking Corg pool, the divergence of the flux remained nearly unchanged at every offshore distance. This suggests that including non-sinking Corg in the model increases the lateral flux, potentially resulting in an increased accumulation and recycling of Corg in the euphotic layer of the open waters, especially at large distances from the coast. However, it does not critically modify our understanding of the impact of the offshore flux on the biological activity of the deep open waters, and may therefore have a minor impact on the resulting regional pattern of heterotrophy. We refer the reader to Lovecchio et al., (2017, 2018) for a further discussion of the strengths and weaknesses of quantifying the Corg lateral transport in the absence of a modeled DOC pool."*

**Answers to the specific comments**

*SC1) Line 10. I would say "the interaction of the Cape Blanc filament with the Cape Vert front".*
In order to acknowledge the important role of the Cape Verde front, we have modified the sentence as follows:
*"linked to the persistent, and far reaching Cape Blanc filament and its interaction with the Cape Verde front."*

*SC2) Line 30. Longer time scales associated to global warning should also be considered.*
We have included the following sentence in the mentioned paragraph:
*"Global climate change further affects the dynamics of these systems (Bonino et al., 2019a; Wang et al., 2015)."*

*SC3) Line 34 (and also in Line 458). Frischknecht et al. 20??.*
Thank you, we have connected this typo.

*SC4) Line 98. Please, indicate integration depth here.*
We have added the following sentence to the mentioned paragraph:
*"Throughout this manuscript, we focus on the 100 m depth, which corresponds to the mean depth of the euphotic layer (i.e. the productive layer) in the region of study."*
However, we also highlight that each figure caption in the manuscript already indicates the exact depth of integration of the fluxes for the specific figure, and that in a few cases the depth of integration is, as indicated, the entire water column.

*SC5) Line 101. See general comments above.*
We have included the following sentence in the Methods in order to refer the reader to the relevant paragraph in the manuscript discussion: *"No dissolved organic pool (DOC) is included in the model; we address this shortcoming in the Discussion (subsection 5.2 "Model limitations")."*

*SC6) Lines 196. Add "eddy" after "mesoscale".*
We agree with this comment and have rephrased the title of subsection 4.2 to: *"Seasonality of the eddy and filament offshore transport"*.

*SC7) Line 400. Comparison with the recent modeling study by Santana-Falcón et al. (2020) should be included here.*
We have included the following paragraph to the discussion:
*"A recent modeling-based study by Santana-Falcón et al., (2020) focuses on the offshore transport of Corg by the Cape Ghirand Cape Juby filaments, inferring an annual net filament flux in the range of between 9.5 and 24.3 kgC s−1 deg−1 for Cape Ghir, and of between 17.1 and 104.9 kgC s−1deg−1for Cape Juby. These high-end estimates of the filament offshore flux include DOC, which accounts for as much as 90% of the transported Corg in Santana-Falcón et al., (2020). Since our NPZD model does not explicitly include a DOC pool (see also "Model limitations" below), this suggests that the filament transport may be even more relevant when DOC is considered"*
Please note that the choice of units used in this new paragraph allows for a better comparison with the papers mentioned in the preceding paragraph.

*SC8) Lines 433-442. As indicated above, the issue of the contribution of deltaDOC to lateral Corg export is very relevant for the manuscript, should be introduced earlier and discussed in more detail. Note also that although deltaDOC is a small fraction of the total DOC it represents from 3 to 20 times the concentration of POC. Furthermore, it is also likely that the deltaDOC transported offshore accumulates in the adjacent oligotrophic gyre because of nutrient limitation to heterotrophic activity rather than the refractory nature of these materials (e.g. Hansell et al., 2009).*
We thank Anonymous Referee nr.1 for this important remark and refer again to our answer to MC3 where we discussed the changes we made in order to address this point more in detail throughout the manuscript.

*SC9) Figures 2 and 3 are somewhat redundant. I would erase Figure 2.*
We disagree on the redundancy of Figure 2, as we believe that having a figure showing the seasonal pattern of the total offshore transport is essential to showing how the overall pattern of Corg lateral relocation changes by season and by latitude. Figure 3, on the other side, only focuses on the mesoscale contribution to the total flux shown in Figure 2. Therefore, Figure 2 provides a comprehensive picture of the total flux that is preliminary to the discussion of the impact of the total flux on the offshore waters, while Figure 3 breaks down the total flux into the physical drivers and is preliminary to the detailed discussion of the important role of the small scale transport.

*SC10) Figure 4. Please, add the non-mesoscale flux contribution too.*
Thank you for this suggestion. We have updated Figure 4 including the non-mesoscale contribution to the total flux (NE-NF flux, gray line). We also updated the spring and summer subplots due to having mistakenly used those for the north sub-region rather than those for the EBUS as a whole. Although this doesn't affect our conclusions, we took advantage of these changes to improve subsection 4.2. We also added a few more sentences regarding the eddy contribution by subregion in subsection 4.3 in order to highlight the different role of the eddies in the two subregions while discussing the bar plots of Figure 6. We are including the new figure and the list of changes to the text below. We have also updated the caption of Figure 4 accordingly.

[Figure]

**Figure 4:** *Seasonality of eddy, filament and non-eddy-non-filament contribution to the total offshore transport of Corg over CanUS integrated throughout the first 100 m depth. The plots show the percentage of the total offshore Corg flux represented by the filament flux (FIL), the cyclonic eddy flux (CE), the anticyclonic eddy flux (AE) and the non-eddy-non-filament flux (NE-NF). Percentages can go above 100% when one flux component is negative, i.e. directed onshore. Subpanels: a) spring (MAM), b) summer (JJA), c) fall (SON), d) winter (DJF). Please, note that the y-axis scale of SON differs from the other subpanels.*

The new dedicated paragraph in subsection 4.2 reads as:

*The CE flux contribution constitutes the largest mesoscale eddy contribution to the total flux in all seasons (Figure 4). Over the CanUS as a whole, the CE flux contributes between 15% and 25% to the total Corg flux, with a contribution that peaks in winter, likely due to a reduced non-mesoscale and filament transport. The weaker AE flux has maximum contribution in the first 500 km from the coast, and its flux share peaks at 15% of the total offshore transport in spring. In fall, the AE flux is slightly negative, i.e. anticyclones weakly recirculate the Corg towards the coast in this season, likely due to stirring.*

The new sentences included in the discussions of the bar plots in subsection 4.3 read as:

*In the offshore NSR, however, eddies (especially CE) contribute importantly to the total flux, with a share that ranges between 30% and 40% of the offshore transport, significantly more than in the CanUS as a whole.*

*Differently from what we find in the NSR, AE have a significant positive contribution to the offshore transport in the nearshore CSR, especially in spring, when the upwelling is strongest south of Cape Blanc.*

SC11) *Figure 10. Change "NCO" by "NCP" in caption.*
Thank you, we have corrected this typo.

---

## Author Comment (AC2) · 3 Mar 2021

**Referee nr.2 Takeyoshi Nagai**

We thank Dr Takeyoshi Nagai (Referee nr.2) for the time spent on reviewing our manuscript and for their useful input and corrections. Following the Referee's comments, we have better clarified our conclusions, stating the processes that drive the latitudinal variations in the seasonality of the offshore flux in the Canary Upwelling System. We have also added a short explanation of the negative contribution of filaments in the text. Further, we have gone through the suggested list of typing errors and corrected them, and have scanned the entire manuscript again in order to spot and amend any additional typo.

We include our detailed answers to their main and specific comments below. We copied in blue the Referee's comments, and included our answers in black. Changes to the manuscript text are highlighted in italics

**Answers to the main comments**

*MC1)   Although the authors mentioned several reasons of the less seasonality in the central sub region, such as the convergence of CC and CUC, and the Cape Blanc filament, the dominant mechanism of this is not clearly written in the conclusion and abstract in the manuscript. I think that the novel finding of this paper is the seasonality variation along the latitude. Therefore, it is important to convey the concise and clear statement for the mechanism, which drives the seasonality variations both in the abstract and the conclusion.*

We appreciate this comment which allowed us to revise and improve our Conclusions. In order to clarify the drivers of the latitudinal and seasonal variations of the Corg offshore flux, our new conclusions read as follows:

*The CanUS is characterized by the most intense seasonal fluctuations in physical and biogeochemical fluxes among all EBUS (Chavez and Messié, 2009) and by an offshore transport of Corg that determines Corg availability in the adjacent open waters. Upwelling processes, driven by the surface alongshore winds, play a crucial role in determining the seasonal variations and latitudinal changes of the nearshore Corg fluxes.*

*Our results show that, in the CanUS, small scale upwelling filaments dominate the total flux at 100 km from the coast in each season and at all latitudes, and are therefore responsible for the seasonal fluctuations of the Corg offshore flux in the nearshore. A combination of eddy and non-mesoscale lateral fluxes extends the Corg transport farther away from the coast. The eddy flux, driven by cyclones, has a maximum offshore extension in spring and summer and maximum relative contribution in winter during moderate upwelling. In the northern subregion, eddies contribute up to 40 % to the total flux offshore of 500km. Anticyclones contribute significantly to the Corg transport only in the nearshore 100 km of the central CanUS in spring, i.e. when the upwelling is maximum south of Cape Blanc. Additional seasonal variability in the alongshore displacement of Corg depicts a dynamic and fully 3D biological pump.*

*Our results also highlight that temporal variations in nearshore processes such as upwelling, production and coastal circulation determine analogous temporal variations in the magnitude and spatial extension of the Corg offshore flux, which is an essential component of the coastal-open ocean biological coupling. In the northern CanUS, the total upwelling shows strong seasonal variability, peaking in summer due to the coastal upwelling. In the central CanUS, the total upwelling shows moderate seasonal variability, peaking in summer and spring due to a combination of coastal upwelling and Ekman pumping. This is reflected in the seasonality of the Corg offshore flux, which in turn contributes to modulate the seasonal changes in the trophic state of the open waters at several hundreds of kilometers off the north-western African coast, albeit with some temporal delay. This delay is shorter than a season only in the nearshore range covered by the intense filament transport. As the dominant scales of temporal variability differ for different EBUS (Chavez and Messié, 2009; Frischknecht et al., 2018), further studies are needed to investigate the repercussions for the open ocean biological activity in regions others than the CanUS.*

**Answers to specific comments**

*SC1) L12 "every season season" Remove one of "season"s.*

Thank you very much, we have corrected this typo.

*SC2) L34 "Frischknecht et al." Publication year is missing.*

We have included the publication year.

*SC3) L55 "N)," Remove a parenthesis. L107 "17âU ¸eN to 24.5â ˚ U ¸eN]" Remove "]". ˚*

Thank you, we have removed it.

*SC4) L108 "This correspond to" should be "This corresponds to"*

We have corrected it.

*SC5) L199 "In every season the filament flux is dominated by a negative (and therefore offshore-directed) Corg flux, with only a minor contribution to the onshore Corg recirculation." Why is the filament flux dominated only by offshore flux, despite that the filament has offshoreward and onshoreward currents on its northern and southern parts, respectively?*

We thank the referee for this very interesting comment.
The filament flux is strongly dominated by a net offshore flux due to the asymmetry in the offshore/onshore velocities associated to, respectively, the northern/southern edges of these structures. The northern side of filaments, in fact, is typically characterized by a faster flow. This asymmetry has been previously observed and documented in several studies, such as, for example:

Álvarez-Salgado et al. 2001: https://doi.org/10.1016/S0079-6611(01)00073-8

Sánchez et al. 2008: https://doi.org/10.1029/2007JC004430

Furthermore, part of the coastal organic matter transported offshore by the filaments may sink and/or be downwelled below the euphotic layer before being recirculated onshore by the southern edge of the structure, therefore resulting in a weaker onshore flux in the first 100m of depth, which is the focus of our study.

As a consequence, at latitudes characterized by an abundant number of filaments that form along the coast, as well as in regions in which the north-south oscillation of the filament structure is remarkable, the intense offshore flux associated with the northern edge of filaments exceeds and cancels the weaker onshore recirculation.

In fact, Figure 3 of the present manuscript (subplots a,c,e,g) does show a weak onshore recirculation associated to the filaments in those regions where these structures are found at a sufficient distance from each other. One clear example is the onshore recirculation in the filament flux located about 100 km south of the Canary Archipelago. This onshore recirculation is likely generated by the southern edge of the intense Cape Juby filament, which is well spaced and separated from the following filament forming along the coast moving south. Analogously, at the northernmost edge of the analysis domain, just north of the Canary Archipelago, the southern edge of the Cape Ghir filament is also characterized by onshore recirculation of Corg.

At southern latitudes, instead, where filaments form densely along the coast, this weak positive signature in the offshore flux gets averaged out.

In order to clarify this point, we have added the following sentence to subsection 4.2:

*"This is due to the faster offshore flow that characterizes the northern edge of a filament compared to the southern onshore filament transport (Álvarez-Salgado et al., 2001; Sánchez et al., 2008), combined with the fact that a portion of the Corg transported offshore may be exported below the euphotic layer before being recirculated onshore."*

*SC6) L247 "In Winter and spring. . ." Uncapitalize "W".*

Thank you, we have corrected it.

*SC7) L306 "offhore" should be "offshore".*

Idem.

*SC8) L339 "and Total upwelling" "T" should be lowercase.*

Idem.

*SC9) L340 "the ratio between offshore transport at 100 km offshore and NCP in the first 100 km from the coast is higher in the CSR in all seasons" Is it the ratio of offshore transport at 100 km offshore to NCP in the first 100 km from the coast, or the other way around?*

We have rephrased the above sentence in: *"the ratio of offshore transport at 100 km offshore to NCP in the first 100 km from the coast is higher in the CSR in all seasons"*

*SC10) L342 ". . .results more pronounced along. . ." Pronounced effects?*

We have rephrased this sentence as:

*"However, the seasonal variation of the upwelling processes is more pronounced in the NRS"*

*SC11) L342 "NRS" Is it "NSR"?*

We have checked. Yes, it is NSR.

*SC12) L345 A comma is needed after "In the NSR".*

Thank you, we have added it.

*SC13) Fig. 11 caption "m3s-1" Numbers should be uppercase.*

We have corrected the caption.

*SC14) L355 "Even thought" should be "Even though". "NCP flux". Is it flux or production rate?*

We have corrected it.

*SC15) L359 "still strong in winter (0.22 Sv)" It's better to refer Fig. 10.*

Thank you, we referred the statement to Figure 10b.

*SC16) L380 "model is agrees" should be "model agrees".*

We have corrected it.

*SC17) L386 "HAGEN, 2001" is all capital. Is it okay?*

We have corrected it.

*SC18) L406 "CanCS" This appears here at the first time in the paper without any definition.*

Thank you, we have changed it to CanUS coherently with the rest of the manuscript.

*SC19) L414 "in the annual mean the CanUS" Remove "the" in front of CanUS.*

We have corrected it.

*SC20) L416 "This excess heterotrophy is fueled by the lateral redistribution of Corg in the offshore and alongshore directions combined with the shoaling of the mixed layer depth in the warm seasons, . . ." Before this sentence, the authors mentioned the effects of deepening of the mixed layer, not shoaling. Do you mean that the shoaling of the mixed layer prevents the Corg from being diffused up?*

What we meant here is that the excess heterotrophy observed in the warm seasons offshore is likely driven by a combination of two factors: 1. the Corg redistribution from the coast to the euphotic layer of the open waters that provides extra Corg for remineralization; 2. a lower nutrient input into the open ocean euphotic layer due to a stratified water column, the latter resulting in low primary production.

As a consequence of the low nutrient availability (stratification) and high Corg availability (lateral input) the warm seasons are the most likely to develop a net heterotrophy in the near-surface open waters.

We have rephrased the mentioned sentence as follows:
*"This excess heterotrophy is fueled by the lateral redistribution of Corg in the offshore and alongshore directions combined with lower offshore productivity driven by the shoaling of the mixed layer depth in the warm seasons..."*

*SC21) L441 "We refer the reader to (Lovecchio et al., 2017, 2018) for a discussion. . ." should be "We refer the reader to Lovecchio et al., (2017, 2018) for a discussion".*

We have corrected this typo.

*SC22) L457 "This delay is smaller than a season only in the nearshore range covered by the intense filament transport." Do you mean that "This delay is typically shorter than one season only in the nearshore range covered by the intense filament transport."?*
Thank you, we have substituted "smaller" with "shorter".